# MemoryBench: A Benchmark for Memory and Continual Learning in LLM Systems

**Qingyao Ai**[1]  **Yichen Tang**[1]  **Changyue Wang**[1]  **Jianming Long**[1]  **Weihang Su**[1]  **Yiqun Liu**[1]

## Abstract

Scaling up data, parameters, and test-time computation has been the mainstream methods to improve LLM systems (LLMsys), but their upper bounds are almost reached due to the gradual depletion of high-quality data and marginal gains obtained from larger computational resource consumption. Inspired by the abilities of human and traditional AI systems in learning from practice, constructing memory and continual learning frameworks for LLMsys has become an important and popular research direction in recent literature. Yet, existing benchmarks for LLM memory often focus on evaluating the system on homogeneous reading comprehension tasks with long-form inputs rather than testing their abilities to learn from accumulated user feedback in service time. Therefore, we propose a user feedback simulation framework and a comprehensive benchmark covering multiple domains, languages, and types of tasks to evaluate the continual learning abilities of LLMsys. Experiments show that the effectiveness and efficiency of state-of-the-art baselines are far from satisfying, and we hope this benchmark could pave the way for future studies on LLM memory and optimization algorithms.

## 1. Introduction

Large Language Model (LLM) has been widely considered to be one of the core techniques to achieve Artificial General Intelligence (AGI). By scaling up training data and model parameters, previous studies have shown that LLMs can obtain richer knowledge and stronger abilities in a variety of downstream applications without task-specific fine-tuning. This phenomenon is referred to as the scaling law (Kaplan et al., 2020). Unfortunately, recent experiments show that the scaling law of LLM pre-training has hit the wall as we are running out of high-quality data and the gains obtained from further increasing parameters become marginal. While teaching LLMs to conduct online reasoning with more tokens (i.e., the inference scaling law (Wu et al., 2025d)) can further improve the system's performance, its potential is bounded by the backbone LLMs (Yang et al., 2025b), and how to train LLMs to obtain such reasoning ability itself is a challenging task that require significant engineering efforts (Yang et al., 2025a). Thus, how to continually improve LLM systems (LLMsys) and achieve higher level of intelligence remains to be an unsolved question.

In fact, from a broader perspective, the concept of continual learning is not new to human and traditional AI systems. Well-known examples include human brains and search engines. Neuroscience shows that people's intelligence could continually grow by interacting with the environment even though the number of neurons in their brains is mostly fixed after the age of 25 (Arain et al., 2013). Modern IR systems such as search engines and recommendation systems can continually improve their performance simply by serving and learning from their interactions with online users. By comparing existing LLMsys with human brains and search engines, we can identify two key components behind the ability of continual learning. The first is a mature and stable **memory** architecture. Human brain has well-developed hippocampus that maintains both context (i.e., short-term memory) and historical (i.e., long-term memory) information, while search engines have well-crafted data and algorithm modules to process dynamic corpus and queries. The second is an effective **feedback** analysis and utilization mechanism. Human brain can effectively learn knowledge and obtain experience through trial and error, which would then serve as the foundation for them to develop better strategies and solutions for tasks in the future; Search engines can utilize different behavior analysis and learning algorithms to understand user's feedback on search results, which could be used as features and training data for system optimization (Croft et al., 2010). Therefore, the key to build a LLMsys with continual learning abilities lays in the development of effective memory architectures and feedback utilization methods (Wu et al., 2025b; Shan et al., 2025; Shi et al., 2025).

[1]Department of Computer Science and Technology, Tsinghua University, Beijing, China. Correspondence to: Qingyao Ai <aiqy@tsinghua.edu.cn>.

*Proceedings of the 43rd International Conference on Machine Learning*, Seoul, South Korea. PMLR 306, 2026. Copyright 2026 by the author(s).

Unfortunately, while memory and continual learning have been widely believed to be important and promising research directions, to the best of our knowledge, there is no well-developed benchmark to evaluate the progress of existing algorithms and guide future studies. As discussed in Section 2, existing benchmark on memory for LLMsys mostly focuses on examining LLM system's ability to handle long context data such as user profiles and conversation histories. They often adopt homogeneous tasks to evaluate LLM systems by simply feeding the context to the memory of LLMsys and ask questions accordingly (Zhang et al., 2025d). More importantly, these benchmarks ignore the dynamic nature of continual learning by evaluating different systems mostly in a static manner (Hu et al., 2025; Maharana et al., 2024; Kim et al., 2025; Zhang et al., 2024; Wu et al., 2025a; Tan et al., 2025). They focus on the retrieval of pre-fetched semantic and episodic memory, and do not support the simulation or evaluation of procedural memory built from test-time user feedback. Without feedback, one cannot learn from its or others' success and failure. This makes these benchmarks not suitable for evaluating the continual learning abilities of LLMsys.

In this paper, we propose a comprehensive benchmark named MemoryBench for the evaluation of memory and continual learning in LLMsys. Inspired by neuroscience (Atkinson & Shiffrin, 1968) and IR (Shen et al., 2005; Agichtein et al., 2006) studies, we categorize the needs of memory based on data characteristics (i.e., declarative or procedural) and build a simulation platform to test the ability of LLMsys in memorizing and learning from user feedback. Our goal is to mimic the process that systems continually learn and improve their performance by interacting with users in online services. To this end, we first collect benchmark datasets in multiple domains, multiple languages, and with multiple tasks that have various lengths of inputs and outputs. Then we split each dataset into separate training and testing sets, and simulate explicit (e.g., a sentence describing whether the output is good or not) and implicit (e.g., a click on the "copy" button) user feedback on the training data using a LLM-as-user paradigm. LLMsys is tested on the testing data to evaluate their abilities in utilizing both the initial input context (i.e., the *corpus*) and the feedback logs to achieve better task performance. We tested both the state-of-the-art (SOTA) LLM memory systems and naive baselines such as retrieval augmented generation (RAG). Experimental results show that existing memory systems for LLM are not good at utilizing procedural knowledge to improve their performance. They are neither effective nor efficient enough to continually learn from user feedback. More advanced parametric and non-parametric algorithms are still needed in order to enable LLM systems to conduct effective continual learning. For reproducibility, all data processing scripts, user simulators, evaluation pipelines, and tested baselines

are open-sourced[1]. We also release simulated user feedback logs for open-sourced LLMs to support off-policy learning experiments[23]. We hope this could pave the way for future studies on memory and continual learning for LLMsys.

## 2. Methodology

### 2.1. Preliminary and Taxonomy

MemoryBench focuses on evaluating the continual learning ability of LLMsys in memory and feedback utilization. Formally, a standard LLMsys consists of two parts: a parametric memory $\theta$ (e.g., the parameters of the LLM) and a non-parametric memory $M$ (e.g., external documents or databases storing the system's knowledge in text). Let $Q$ be the set of tasks issued by users, then a LLMsys can generate a set of responses as $f(\theta, M, Q)$. Let $t$ be the time step and $s(q_t, f(\theta_t, M_t, q_t))$ be the feedback generated by the user (or the environment) based on the response of LLMsys to the task query $q_t$, then the goal of continual learning is to find $\theta_t$ and $M_t$ based on $S_{t-1} = \{s(q_i, f(\theta_i, M_i, q_i))\}_{i=1}^{t-1}$ so that the loss on the next query $q_t$, which we define as $l(q_t, f(\theta_t, M_t, q_t))$, could be minimized. We refer to $S_t$ as the feedback logs, and the key question of continual learning is how to improve LLMsys's performance on future tasks based on its memory $M$ and feedback logs $S$.

The concept of memory in intelligent systems has been extensively studied in the field of neural science, social science, and computer science. For example, previous studies have developed different taxonomies for memory based on what they store (e.g., facts, events, feelings, etc.) and how they are managed (e.g., short-term, long-term, etc.) (Atkinson & Shiffrin, 1968). In this paper, our goal is to evaluate different LLMsys with or without memory but not developing new architectures or algorithms. Thus, inspired by previous studies (Cohen & Bacdayan, 1994; Tulving & Markowitsch, 1998), we analyze memory based on the knowledge they store and categorize them into two types: *Declarative Memory* and *Procedural Memory*. Declarative memory refers to factual knowledge independent to tasks. Depending on whether the knowledge is related to users, we further categorize it into *Semantic Memory* and *Episodic Memory*. We refer to semantic memory as data and knowledge independent to users, such as textbook, wikipedia, or other factual information collected from the world, etc. We refer to episodic memory as memory that is user dependent, such as conversational history, personal profiles, or other user-specific information, etc. In contrast to declarative memory, procedural memory focuses on non-factual knowledge re-

---

[1] https://github.com/THUIR/MemoryBench
[2] https://huggingface.co/datasets/THUIR/MemoryBench
[3] https://huggingface.co/datasets/THUIR/MemoryBench-Full

*Table 1.* Benchmarks Categorization and Comparison. Dec., Pro., Sem., Epi., Exp., Imp., Ver., Act. are abbreviations for Declarative, Procedural, Semantic, Episodic, Explicit, Implicit, Verbal, and Action. LiSo is the acronym of Long-input-Short-output, etc.

| Name | Task | Memory | | | Feedback | | | Number of Cases |
|---|---|---|---|---|---|---|---|---|
| | | Dec. | | Pro. | Exp. | | Imp. | |
| | | Sem. | Epi. | | Ver. | Act. | | |
| Hu et al. (2025) | LiSo | ✓ | ✓ | × | × | × | × | 2k |
| Wu et al. (2025a) | LiSo | × | ✓ | × | × | × | × | 0.5k |
| Kim et al. (2025) | LiSo | × | ✓ | × | × | × | × | 3121k |
| Maharana et al. (2024) | LiSo | ✓ | ✓ | × | × | × | × | 7k |
| Du et al. (2024) | LiSo | ✓ | ✓ | × | × | × | × | 8k |
| Zhang et al. (2024) | LiSo | ✓ | ✓ | × | × | × | × | 3k |
| Tan et al. (2025) | LiSo | ✓ | ✓ | × | × | × | × | 53k |
| MemoryBench (ours) | LiSo; LiLo; SiLo; SiSo | ✓ | ✓ | ✓ | ✓ | ✓ | ✓ | 20k |

lated to the execution of tasks, such as the workflow of a task, the reward of a solution, etc. For instance, user feedback and behavior logs that contain important information about how the system could improve its performance in the future can be considered as important resources for the procedural memory of LLMsys. An illustrative example is shown in Figure 1 to better show the relationships and differences between different types of memory information.

In contrast to memory, the taxonomy of user feedback is not well explored in existing LLM literature. While the word "feedback" is not new for LLM, most existing works use it as a general term for training and testing reward rather than user behavior signals collected in LLM services (Ouyang et al., 2022). Inspired by user studies in IR (Shen & Zhai, 2005; White et al., 2006), we categorize user feedback for LLMsys into two types: *Explicit Feedback* and *Implicit Feedback*. Explicit feedback refers to user behaviors that directly reflect the quality of the LLMsys's output. For example, a user may conduct multi-turn conversations with a LLMsys by directly telling it what's wrong with its output and how to fix it, which we refer to as *verbal* feedback. A user may also tell the system whether its response is satisfying by simply clicking the "like" or "dislike" button widely provided in chatbot user interfaces, which we refer to as *action* feedback. An example can be found in Figure 1. In contrast, implicit feedback denotes user behavior actions generated during the use of LLM services, but not for the purpose of judging the quality of LLM responses. Examples include clicking the "copy" button, closing a session after seeing the response, or starting a new session with a similar but refined prompt. These behaviors could be useful for optimizing the system, but may contain noise and need significant analysis in order to extract reliable reward signals. Based on this taxonomy, we analyze and categorize existing benchmarks on LLM agent and memory in Table 1. To the best of our knowledge, MemoryBench is the first benchmark that provides all types of memory and feedback data. Also, previous benchmarks mainly focus on testing LLMsys

on answering factual questions where the answers can be extracted from the given context (i.e., the *corpus*), which are essentially reading comprehension tasks with long input and short output. In contrast, in MemoryBench, we include diverse tasks from multiple domains that have various lengths of input and output (details in A.1).

### 2.2. Architecture

Shown in Figure 2, the framework consists of three modules: *Task Provider*, *User Simulator*, and *Performance Monitor*.

**Task Provider** collects the query $q$, the user/task context $c$ (i.e., the *corpus*), and the evaluation metadata $v$ (e.g., the ground truth, evaluation criteria, see details in Section 2.3 and A.1) from the training and testing data of each dataset. In off-policy experiment settings (see details in Section 3 and A.3), it also collects the pre-fetched feedback logs $S$ for each task. The training cases are fed into LLMsys and user simulator to generate responses and corresponding feedback. The testing cases are fed into LLMsys and performance monitor to evaluate the system performance. Here, the user/task context $c$ and the feedback logs $S$ are the main resources for the LLMsys to build its declarative and procedural memory.

**User Simulator** aims to mimic the way how real users interact with LLMsys. Based on the training queries and evaluation metadata provided by the task provider, it simulates human-like feedback to LLMsys's responses. This includes both explicit and implicit feedback in verbal text responses or actions described in Section 2.1. We implement the user simulator with a LLM-as-user paradigm, and it could interact with the LLMsys for multiple turns per session. The number of turns varies according to the experiment setting, and the corresponding feedback and dialogs would form the user behavior logs $S$ that could further be used in the procedural memory of LLMsys.

**Performance Monitor** is designed to monitor the performance of LLMsys on the test data. It examines whether LLMsys can improve their task performance through in-

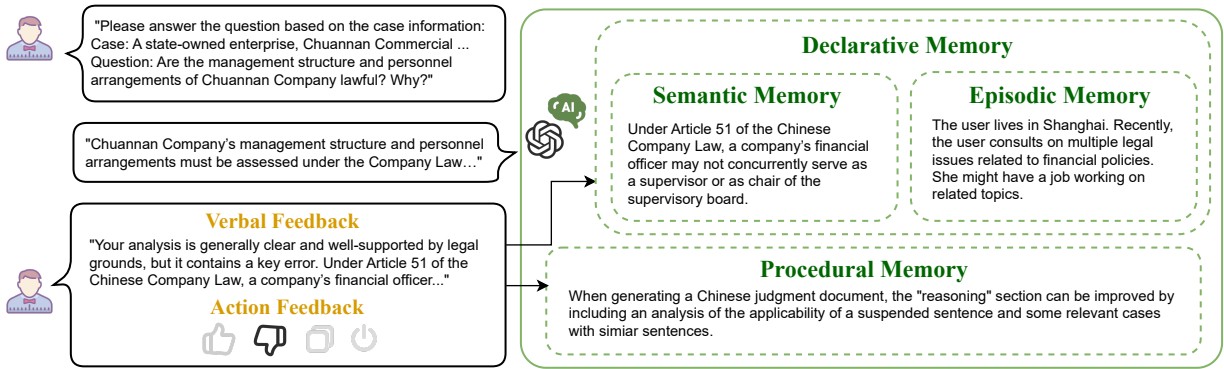

*Figure 1.* An illustrative example of different types of memory and user feedback.

teracting with the simulated users. Specifically, we adopt the native metrics used by the original datasets for evaluation. Because different tasks may involve multiple and diverse evaluation metrics, we implement a LLM-as-judge paradigm (Li et al., 2024b) to merge multiple metrics into one performance score per task case. We aggregate multiple task scores by first conducting a min-max normalization or computing the z-score of all tested LLMsys and test cases within each dataset, and then averaging the performance score of a specific LLMsys on test cases from all datasets to compute the final score for this LLMsys. Note that the performance monitor evaluates LLMsys only on the test data, which are different from the training data fed by the task provider to the user simulator.

### 2.3. Data Preparation

To analyze LLMsys's ability in utilizing declarative and procedural memory in heterogeneous tasks, we collect 11 public datasets across three domains, four types of task formats, and two languages. Specifically, they are Locomo (Maharana et al., 2024), DialSim (Kim et al., 2025), LexEval (Li et al., 2024a), JuDGE (Su et al., 2025), IdeaBench (Guo et al., 2025b), LimitGen-Syn (Xu et al., 2025b), Writing-Prompts (Fan et al., 2018), HelloBench (Que et al., 2024), WritingBench (Wu et al., 2025c), NF-Cats (Bolotova et al., 2022), and SciTechNews (Cardenas et al., 2023). As shown in Table 3, A.1, these datasets cover open-domain, legal, and academic data, and their tasks exhibit wide variations in input and output formats and lengths. We believe that these datasets can closely reflect many daily usage scenarios for LLMsys, which are suitable for the generation and collection of user feedback logs. For datasets that span multiple tasks and domains (e.g., WritingBench), we further divide them into subsets based on our categorization taxonomy. For all datasets, we unify their representations and format each task case as $(q, v, c)$, where $q$ denotes the input query or instruction submitted by the user, $v$ represents the metadata needed for evaluation (e.g., datasets such as

JuDGE and WritingBench rely on either golden answers or multiple hand-crafted criteria to evaluate LLMsys's responses), and $c$ is the task context, i.e., the *corpus*, needed to answer the query. Here, $c$ is optional as it depends on the task and dataset. For example, Locomo and DialSim require LLMsys to answer questions based on user's previous dialogs, and these dialogs are considered as $c$ for the task. Also, depending on the tasks, $c$ could be user-dependent or user-independent. For simplicity, we do not differentiate it explicitly and let the LLMsys decide how to process the task context. The design of $v$ is closely related to the main test metrics used by each dataset. For datasets that have multiple metrics (e.g., IdeaBench, JuDGE, and SciTechNews), we adopt a LLM-as-judge paradigm and merge those metrics into one. Please refer to A.1 for more details.

### 2.4. Feedback Simulation

A core component of MemoryBench is to simulate realistic user feedback at scale, which is essential for evaluating the continual learning capabilities of LLMsys. To this end, we implement a hybrid feedback simulation mechanism that combines a LLM-as-user paradigm with a two-stage programmable action simulator. Specifically, for tasks that have a verifiable ground truth (e.g., information extraction, fact-checking), the LLMsys's output is directly compared against the dataset's ground truth using objective metrics (e.g., F1-score, accuracy). The resulting score is then mapped to pre-defined feedback templates that generate corresponding explicit and implicit signals (e.g., a high score yields a positive response or a "like" action, see details in A.2). For open-ended and subjective tasks, we employ an LLM to simulate a user's cognitive and behavioral responses. To ensure the relevance and reliability of the feedback, the simulator is instantiated with a specific user persona, which defines its background, level of domain expertise, and the evaluation criteria it should apply (if provided by the dataset). Then, the simulator is prompted to produce a response to simulate user feedback based on our taxonomy:

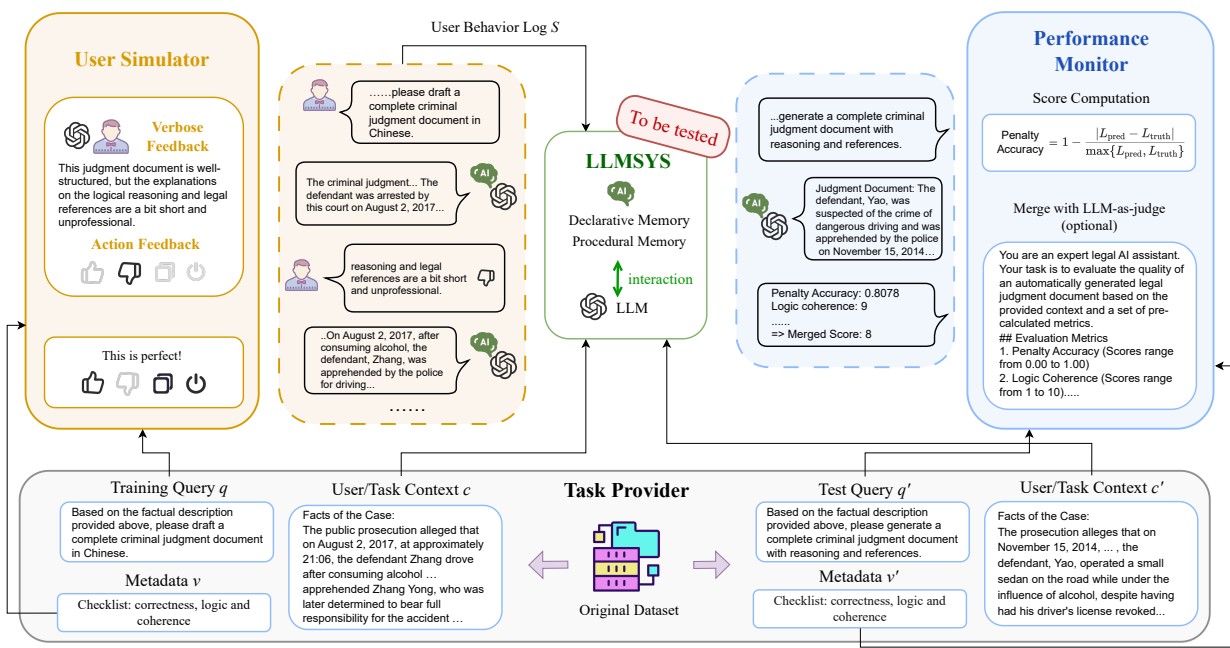

*Figure 2.* The framework of MemoryBench includes: (1) **Task Provider** collects the query $q$, context $c$, and evaluation metadata $v$ for each task case; (2) **User Simulator** simulates human-like feedback and interactions to the LLMsys (which serves as the feedback logs $S$) on training data; (3) **Performance Monitor** evaluate the actual performance of LLMsys on the test data.

- **Verbal Feedback**: The simulator generates a (1) natural language critique that analyzes the quality and relevance of the LLMsys's response, and (2) a subsequent conversational turn (response) if the user's intent is to continue the interaction. This provides an explicit, verbal signal about the LLMsys's response quality.

- **Action Feedback**: The simulator predicts a user satisfaction score for the LLMsys's response and generates explicit and implicit actions such as "like", "dislike", or "copy" based on mapping functions. The functions are task-dependent and programmable so that the generated actions could follow diverse user behavior patterns. We design the mapping functions following the observations obtained from recent studies on user behaviors on LLM services (Shuster et al., 2022; Chatterji et al., 2025). More details can be found in A.2.

To verify the quality of our LLM-as-user paradigm in terms of generating high-quality verbal feedback, we have conducted human annotation experiments to compare human-written feedback with those generated in MemoryBench. Results show that it is difficult for annotators to distinguish human feedback with our simulated feedback (i.e., either the annotators find our simulated feedback is more human-like or their kappa agreements on which feedback is more human-like is low). More details can be found in A.2.5.

## 3. Experiments

### 3.1. Baseline Categorization

To demonstrate how to use MemoryBench, we implemented and tested simple baselines as well as a couple of SOTA memory-based LLMsys, which are naive retrieval augmented generation (RAG) with BM25 and Qwen3-Embedding-0.6B (Zhang et al., 2025c) as the retrievers, **A-Mem** (Xu et al., 2025a), **Mem0** (Chhikara et al., 2025), and **MemoryOS** (Kang et al., 2025). We use the same backbone LLM (i.e., Qwen3-8B, but we have also tested Qwen3-32B and Mistral3.2-24B as shown in Table 15 and Table 22 in A.3) for all baselines and refer to the naive method that directly answers each query with the backbone LLM as **Vanilla**. For memory-based LLMsys, we directly adopt their original code and design from their papers to manage input corpus and feedback logs. For RAG methods, we store the corpus directly and tested two methods to process the feedback logs: forming a single document with the whole dialog in each session, or storing each message in a session as separate documents. We refer to the baselines using BM25 and Qwen3-Embedding-0.6B with the Session documents and Message documents as **BM25-S**, **BM25-M**, **Embed-S**, and **Embed-M**, respectively. For action-based feedback such as "like" and "copy", we also implemented a supervised fine-tuning baselines (SFT) and reported its performance in A.3.9. When the input context of a case is longer than the input limit of a baseline, we simply cut the

context to fit. More details can be found in A.3.

## 3.2. Experimental Setup

*Data Partition.* We build the testbeds of our experiments by grouping data based on their domains and task formats. The goal is to evaluate the baselines' generalizability across tasks within the same domain or across domains with similar task formats. Specifically, we have 7 types of data partitions as **Open-Domain**, **Legal**, **Academic** for domains and **LiSo**, **SiLo**, **LiLo**, **SiSo** for task formats (details can be found in A.1). Within each partition, we evenly sampled the same amount of cases from each dataset (unless the size of the dataset is smaller than our sample size) randomly, split them into training data and test data with ratio 4:1, and merge them across all datasets to form the final training set and test set of the partition. Feedback logs are generated only on the training set so that no test information is leaked.

*Evaluation Metrics.* MemoryBench follows the original evaluation metrics of each dataset. Many of these datasets (e.g., JuDGE, LexEval, IdeaBench, SciTechNews) have human-crafted ground truth and official evaluation protocols that are used for comparison. These gold standards often represent near-human-optimal responses and provide important information about the performance ceiling for human and LLMsys on each task (please refer to the dataset references for details). Because many datasets in MemoryBench have multiple evaluation criteria in their official settings, for simplicity and better visualization, we employ a LLM-as-Judge paradigm to merge these metrics into a single ranging from 1 to 10. We present the prompts used for this aggregation in A.1. Then, for the overall evaluation of a task or domain, we apply min-max normalization or compute the z score to all results within each dataset before calculating the average performance of all test cases. We also report the original metric performance on each dataset in our off-policy experiments in Table 12, A.3.

*On-policy and Off-policy Settings.* In each experiment, we first feed all the initial task context (i.e., the corpus for each training case) to a LLMsys, and then explore two settings to evaluate it. The first one is **off-policy** learning. We first run the backbone LLM of the LLMsys on all training cases and simulate 1 dialog session per each case to form the feedback logs. At each time step, we feed one batch of training cases and corresponding feedback sessions (one per each case) to the LLMsys. The second one is an **on-policy** learning setting. In each time step, we feed a batch of randomly sampled training cases to the LLMsys and collect feedback. The feedback logs are then fed to the LLMsys so that it can update its memory or parameters before the next time step. However, as discussed in Section 3.3 and A.3.8, some memory-based LLMsys are too slow that running on-policy experiments for them on all datasets would take days or even

weeks. Therefore, we only reported the on-policy results for LLMsys that can finish the tasks within reasonable time (i.e., less than a day) in A.3. In MemoryBench, we simulate user feedback directly with a LLM-as-user paradigm using a strong LLM (i.e., Qwen-32B if not explicitly mentioned otherwise, but we also provide the feedback by Mistral3.2-24B) and the evaluation metadata of each task. More details can be found in A.2 and A.3. For simplicity, if not explicitly stated otherwise, all experiments reported in the main content of this paper are conducted in the off-policy setting with explicit verbal feedback only. We also provide more experiment results on on-policy settings and other types of feedback (e.g., explicit actions and implicit actions such as clicks on "like" and "copy" buttons) in A.3.

## 3.3. Results and Discussions

Our experiments on Memorybench mainly focus on three questions: (1) Whether our simulation framework can provide reasonable user feedback that helps LLMsys improve their task performance; (2) How do different LLMsys perform in utilizing user feedback for continual learning; and (3) What are major limitations and potential future research directions for memory-based LLMsys.

**Usefulness of simulated user feedback.** Figure 3 presents the overall performance results (off-policy settings, explicit verbal feedback only, more detailed and on-policy results can be found in A.3) of all the baselines tested in our experiments on different partitions of MemoryBench. Note that the results for Mem0 is missing on Open-Domain and LiSo because the context lengths of those task data are so long that Mem0 cannot process in its memory and produce responses in reasonable time (more details can be found in A.3.8). While the best performing methods may vary across different domains and task formats, LLMsys with memory support mostly outperformed the Vanilla method that uses the backbone LLM without memory. To further validate that the reliability of the simulated user feedback, we conducted experiments comparing the performance of the backbone LLM (i.e., Vanilla) on each task case with and without feedback (results can be found in A.3.5). Our experiments show that the LLM-as-user paradigm could produce reasonably accurate feedback to LLMsys's responses that are indeed useful for LLMsys to improve their performance. Yet, in Figure 3, the performance of Vanilla on SiLO is better than all memory-based LLMsys, which indicates that the effectiveness of existing LLMsys utilizing user feedback could vary significantly across tasks with different formats.

**Comparison of different LLM-based LLMsys.** Despite of the rising popularity of memory-based LLMsys, our experiments on MemoryBench show that the SOTA memory-based LLMsys are still unsatisfying. As shown in Figure 3,

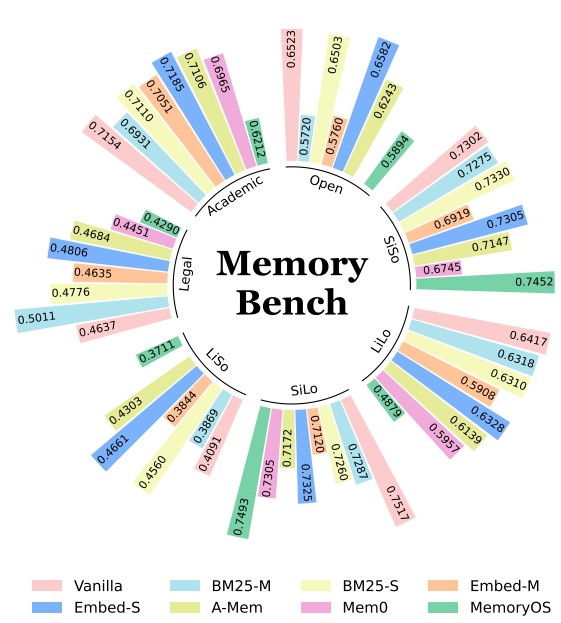

*Figure 3.* The off-policy experimental results on MemoryBench with min-max normalization based metric score merge.

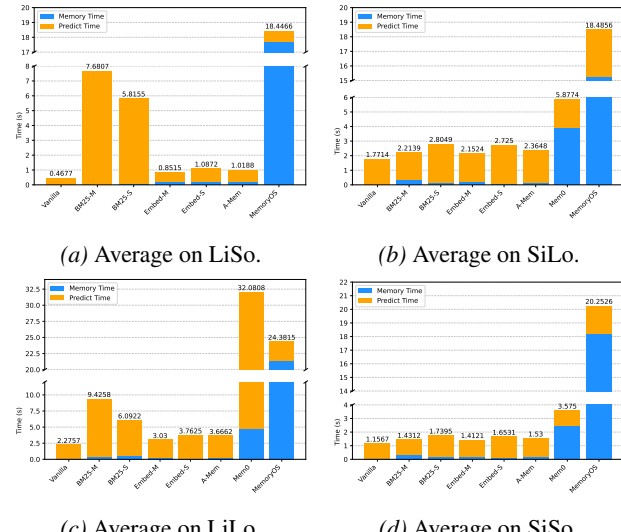

*(a) Average on LiSo.*   *(b) Average on SiLo.*

*(c) Average on LiLo.*   *(d) Average on SiSo.*

*Figure 4.* Time consumption of different LLMsys in off-policy experiments on four task-format-base partitions of MemoryBench. The bars show the average time per case for memory operations (Memory Time, in blue) and prediction on the test set (Predict Time, in orange), measured in seconds. Note that the construction of memory indexes or retrieval indexes are also considered as a part of the Memory Time. More details can be found in A.3.8.

none of the advanced memory-based LLMsys (i.e., A-Mem, Mem0, or MemoryOS) can consistently outperform RAG baselines that simply use all task context and feedback logs as retrieval corpus. To understand whether this phenomenon is related to the types of feedback signals we collected, we also conducted experiments with the simulated action feedback such as "like" and "copy", where we observed similar results (more details in A.3.9). This is contradicting to the good results reported by former studies (Xu et al., 2025a; Chhikara et al., 2025; Kang et al., 2025). After a careful examination of our experimental design and previous studies, we find several differences in our experimental design. First, previous studies only tested their system on reading comprehension tasks that involve long input context and require short answers (e.g., Locomo). They manually tailored their methods to handle such tasks using special chunking, summarization, and hierarchical memory-clustering techniques to handle long input context. In contrast, the RAG baselines used in our paper simply build their retrieval corpus by splitting the context by dialogs or messages without any task-specific design. If we only look at the models' performance on Locomo in our experiments, existing memory-based methods indeed outperformed naive RAG baselines with BM25 (see Table 12 in A.3). However, Locomo is only one of the datasets tested in MemoryBench and many datasets have different input and output formats with it. This indicates that the generalizability of these SOTA memory-based LLMsys is limited. Second, MemoryBench provides both the current task context and the feedback logs from previous tasks to LLMsys. This makes

it significantly different from previous benchmarks where only the current task context is provided. As discussed in Section 2.1, the task context provided by previous benchmarks are usually factual information independent to how the system performs with respect to the task in the history (i.e., declarative memory). The feedback logs provided in MemoryBench are non-factual information describing how the system performed and what it has done well or badly in historical tasks (i.e., procedural memory). Existing memory-based LLM systems such as Mem0 and MemoryOS simply treat all inputs as declarative memory and develop memory mechanisms accordingly. This limits their ability to understand and utilize procedural memory, i.e., the feedback logs in MemoryBench. While RAG baselines do not have sophisticated mechanism to analyze and utilize different types of task context and user feedback, their retrieval systems (i.e., term-based index systems or vector retrieval engines) are strong in handling large-scale data collection and retrieving relevant information efficiently. Thus, their performance are comparable to many advanced memory methods on MemoryBench.

**Limitations of existing memory-based LLMsys.** In our experiments, we notice that both the effectiveness and the efficiency of existing memory-based LLMsys are far from perfect, especially for continual learning with increasing amount of user feedback. First, we are surprised to find that the time of existing memory-based methods used to memorize input context and conducting online inference

are extremely long and varied inconsistently across tasks. Figure 4 shows the Memory Time (i.e., the time a LLMsys needs to process the behavior logs and the task context of one test case, including the time of index construction in RAG methods) and the Predict Time (i.e., the time a LLMsys needs to retrieve memory and generate a response to a test case) of different LLMsys on tasks with different formats. As shown in the figure, Mem0 has small memory time on tasks with long inputs (i.e., LiSo, LiLo), but longer time on tasks with short inputs (i.e., SiLo, SiSo). And comparing to other baselines, its inference speed is abnormally large on LiLo. MemoryOS exhibited better consistency in terms of inference time, but it took much longer time in memory construction (mostly longer than 17 seconds per case). A-MEM is the most efficient memory-based LLMsys in our experiments, probably thanks to its simplified memory construction and retrieval process. Yet, its effectiveness is not better than naive RAG baselines in many cases. Also, in our experiments, the performance of LLMsys fluctuated significantly during the training process, especially on Academic and Legal domains (see in A.3.7). This means that existing methods cannot effectively filter and utilize user feedback in continual learning. One possible reason is that, compared to Open domain, feedback logs on vertical domains might be more difficult to interpret as it requires more domain knowledge. Overall, our experiments on MemoryBench show that the memory mechanisms of existing LLMsys are not capable of handling multiple types of declarative and procedural memory at the same time.

## 4. Related Work

**Memory-based LLM systems**. The studies of LLM memory mostly focus on constructing external information management systems that help LLMsys accumulate knowledge, process experience, make decisions, etc. (Zhang et al., 2025d). In general, advanced memory-based LLMsys consists of multiple modules including but not limited to a memory processor that take user-system conversations as input to build memory entries/documents; a memory database that store and indexes memory entries; a memory retriever to retrieve relevant entries given a specific context; and a generator that takes the retrieved memory and current task context to generate the responses. A typical LLM memory system support multiple operations such as writing(Zhong et al., 2024), summarizing (Liu et al., 2023), updating (Xu et al., 2025a), deleting (Park et al., 2023), and retrieval (Packer et al., 2023). It operates its memory database when it interacts with the environments, and utilize relevant memory to better improve its responses. The applications of memory cover many use cases including social simulation (Zhang et al., 2025b), personal assistant (Lu et al., 2023), gaming (Yan et al., 2023), and much more.

**Continual Learning for LLM**. The concept of continual learning, especially those based on user feedback, is old to AI but relatively new to LLM. Most studies in LLM literature focuses on the pre-training (Devlin et al., 2019) and post-training (Achiam et al., 2023) of LLMs based on SFT (Hu et al., 2022), RL (Guo et al., 2025a) or other algorithms. There are many papers on continual learning algorithms that adapt LLM to unseen data distributions (Shi et al., 2024), but not for continually optimizing LLMsys with user feedback. One possible reason is that no commercial LLM service company has released their user logs, and there isn't a comprehensive benchmark that has user feedback for LLMsys. While there has been studies analyzing LLMsys's ability in multi-turn conversations (You et al., 2024), they considered these utterances only as local context and discard them after the system finish each task. To the best of our knowledge, MemoryBench is the first benchmark that can evaluate the continual learning ability of LLMsys on user feedback logs.

**Feedback Simulation**. User feedback simluation has been widely studied in IR for more than fifty years (Balog & Zhai, 2023), particularly on evaluation (Zhang et al., 2017), information seeking activities (Zhang et al., 2025a), search and recommendation behaviors (Craswell et al., 2008), conversational agents (Wang et al., 2024), etc. Traditional user simulation frameworks are often designed for well-developed yet simple user interfaces such as ranked lists (Wang et al., 2013). As the UI designs of LLMsys gradually become homogeneous, there are more studies on user behavior analysis for LLMsys recently (Liu et al., 2024). With the help of advanced simulation algorithms (Wang et al.) and LLM-as-judge frameworks (Li et al., 2024b), it is now possible to simulate user feedback to LLMsys with good quality.

## 5. Conclusion

In this paper, we construct MemoryBench, a new benchmark to evaluate the memory and continual learning abilities of LLMsys. In contrast to existing benchmarks that only focus on tasks with long input context, MemoryBench evaluates whether LLMsys can take advantages of user feedback in service time and construct effective procedural memory to improve their performance continuously. To this end, we simulate user feedback with a LLM-as-user paradigm on tasks across multiple languages, domains, and task formats. Experiments show that existing memory-based LLMsys are far from perfect to utilize these feedback in effective and efficient way. While we acknowledge that the simulation framework used by MemoryBench may introduce systematic biases to the evaluation process, it is still clear that there is still a long way before we build a LLMsys that has actual continual learning abilities. We hope MemoryBench could serve as one of the initial steps to facilitate future studies.

## Impact Statement

The MemoryBench benchmark seeks to advance the field of memory and continual learning for LLMsys. By focusing on the integration of user feedback and diverse tasks, we aims to foster algorithms and systems that can adapt and improve over time by serving people, similar to human and search engines. However, such systems might present ethical challenges, including the risk of reinforcing biases or misinterpretations based on imperfect feedback, leaking user privacy in the optimization and service process. These risks are not significant in Memory-Bench as it is built from public LLM datasets with simulated user feedback, but we suggest researchers and developers to prioritize ethical considerations in their building own systems. This includes implementing rigorous validation processes, ensuring transparency in data usage, and maintaining oversight mechanisms to mitigate potential biases and protect user privacy. To support these ethical goals and ensure scientific integrity, we have open-sourced all data, pre-processing scripts, and baseline implementations at https://huggingface.co/datasets/THUIR/MemoryBench and https://github.com/THUIR/MemoryBench, facilitating the reproducible study of adaptive LLM systems.

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

# A. Appendix

## A.1. Data Preparation and Evaluation Protocols

Table 3 reports the statistical information of the datasets included in MemoryBench. To measure text length, we compute the number of tokens after tokenized by the Qwen3-8B tokenizer. For inputs, we combine the query with the corpus (if available), while for outputs, we use the length of the golden answers. For datasets without golden answers (e.g., HelloBench, IdeaBench, WritingBench), we record the average length that Qwen3-8B actually generates. We set 600 tokens as the boundary for distinguishing between short and long inputs/outputs. In the LLM-as-Judge evaluation, we follow the official prompt templates provided by each dataset. We use the officially released WritingBench-Critic-Model-Qwen-7B[4] for WritingBench, and DeepSeek-V3 for the other datasets.

### A.1.1. DETAILS OF DATA PREPARATION

Here we introduce the detailed descriptions of each datasets used in MemoryBench. A categorization of the datasets based on the taxonomy of memory used in Section 2.1 is shown in Table 2. Specifically,

- **LexEval** (Li et al., 2024a): LexEval is a comprehensive Chinese legal benchmark. We select three subsets labeled 5-1, 5-2, and 5-4, corresponding to Summary Generation, Judicial Analysis Generation, and Open-ended Question Answering. The evaluation metric is the Rouge-L score compared to the golden answers.

- **JuDGE** (Su et al., 2025): JuDGE is a Chinese Legal Judgment Document Generation evaluation dataset. LLMsys need to generate a complete Judgment Document based on the facts of legal cases. We incorporate both the JuDGE training and test sets as a subset of MemoryBench. JuDGE uses a broad range of evaluation metrics, focusing primarily on differences from the golden documents annotated by human experts. These metrics include the similarity of the Reasoning and Judgment Sections (assessed via METEOR and BERTScore), the accuracy of legal references and charges (measured by recall, precision, and F1), and the relative differences in fines and prison sentences.

---

[4] https://huggingface.co/AQuarterMile/WritingBench-Critic-Model-Qwen-7B

- **LimitGen-Syn** (Xu et al., 2025b): LimitGen systematically evaluates how LLMs identify limitations in scientific papers, particularly in AI research. It includes a human-annotated subset (LimitGen-Human) and a synthetic subset (LimitGen-Syn). Because the former requires multiple dialogue sessions to generate answers, and to align with the other MemoryBench datasets, we only choose the LimitGen-Syn subset. We follow the original paper's setup by specifying in the prompt which limitation subtype (e.g., Methodology, Experimental Design) should be recognized, and we instruct LLMsys to generate three limitations in JSON format. Following the original paper, we use an LLM-based rating metric (ranging from 0–5) during evaluation. We first match the limitation subtype at a coarse-grained level, then compare these matches to the ground truth to assign scores, and finally select the highest scoring limitation among the three generated by LLMsys as the final result.

- **WritingBench** (Wu et al., 2025c): WritingBench is a multi-domain benchmark that evaluates writing proficiency in both English and Chinese using dynamic evaluation rules, with the critic model required to produce scores from 1 to 10. We build three MemoryBench subsets from WritingBench:

  - **WritingBench-Politics&Law**: *Politics & Law* task in WritingBench, including government documents generation, legal writing, etc.
  - **WritingBench-Academic&Engineering**: *Academic & Engineering* task in WritingBench, including academic papers generation, patent writing, etc.
  - **WritingBench-Creative&Design**: *Literature & Arts*, *Education,* and *Advertising & Marketing* tasks in WritingBench, involving art, creativity, or design.

- **WritingPrompts** (Fan et al., 2018): WritingPrompts is a story generation dataset, from which we use the first 2000 items in the test set. Evaluation relies on the METEOR score (Banerjee & Lavie, 2005), comparing generated stories to human-written ones.

- **Locomo** (Maharana et al., 2024): Locomo is a dataset designed to evaluate the long-term memory capability of LLMsys. It includes 10 long dialogues, each containing around 200 questions. We treat each dialogue (session-level) as the corpus and each question as a user query. We then calculate the Average F1 score of all answers against the gold answers (denoted in the original paper as "*Answer Prediction (F1) – Overall*").

- **DialSim** (Kim et al., 2025): DialSim aims to evaluate the Long-Term Multi-Party Dialogue Understand-

*Table 2.* The categorization of datasets in MemoryBench in terms of memory types. Declarative, Procedural, Semantic, Episodic are abbreviations for Declarative, Procedural, Semantic (user independent memory), and Episodic memory (user-specific memory).

| Dataset Name | Memory Type | | |
| --- | --- | --- | --- |
| | Declarative | | Procedural |
| | Semantic | Episodic. | |
| LexEval | ✓ | × | × |
| JUDGE | ✓ | × | × |
| LimitGen-Syn | ✓ | × | × |
| WritingBench | ✓ | × | × |
| WritingPrompts | ✓ | × | × |
| Locomo | × | ✓ | × |
| DialSim | ✓ | ✓ | × |
| SciTechNews | ✓ | × | × |
| IdeaBench | ✓ | × | × |
| HelloBench | ✓ | ✓ | × |
| NF-Cats | ✓ | × | × |

ing of LLMsys. We adopt the official v1.1 version of the dataset, and divide its content by the TV show source into three subsets: *friends*, *bigbang*, and *the-office*. Each subset's corpus comprises the complete scripts of the corresponding TV show, and the user queries are drawn from *Fan Quiz*-type questions. We only consider questions that are answerable and with time to ensure alignment between the corpus and the queries. In the official DialSim prompt, the full dialogue history is given directly to the model; however, within MemoryBench, we require each baseline to handle memory automatically. To accommodate this, we minimally modify the original prompt by removing the dialogue history section and leaving everything else unchanged, as follows:

---

**Prompt Template of DialSim for Memory-Bench**

You are { *Chatbot* }, a long-term conversation agent capable of interacting with multiple users.
Based on the [Dialog History] provided, please answer the given [Question].
Note the following points:
1. Your responses should solely rely on the retrieved dialog history. If the information in the dialog history is insufficient to answer the question, you must admit that you don't know the answer.
2. This question is being asked in the context of { *Date* }.

[Question] { *Question* }
[Answer]

---

Due to the large dataset size, we randomly select 1,000 queries from each subset for MemoryBench. The evaluation of DialSim is an LLM-assisted accuracy assessment: first, we use exact match to check whether the predicted answer is identical to the golden answer; if exact matching fails, we then use LLM-as-Judge for further verification.

- **SciTechNews** (Cardenas et al., 2023): SciTechNews is a dataset of Science Journalism for the General Audience. We use its test set and follow the evaluation settings proposed in the original SciTechNews paper and the JRE-L framework (Jiang et al., 2025). Our metrics include two primary dimensions. The first examines how closely the generated journalism aligns with the human-provided golden answers, using Rouge-L and BERTScore-F1 (Zhang et al., 2020) to calculate factual accuracy. The second relies on readability measures, such as CLI (Coleman & Liau, 1975), FKGL (Kincaid et al., 1975), and DCRS (Dale & Chall, 1948), to assess the readability of the generated journalism. A lower readability score indicates that the article is better suited for science news.

- **IdeaBench** (Guo et al., 2025b): IdeaBench is a benchmark that evaluates the generation of research ideas by LLMsys, requiring LLMsys to create new research hypotheses or ideas based on a set of reference abstracts. Its evaluation framework is multifaceted and mainly examines how the generated ideas align with the "real" ideas in the target paper under various quality metrics. The primary metric is the "Insight Score," obtained through a two-step procedure: first, LLM-as-Judge ranks all ideas against user-specified criteria (such as "novelty" and "feasibility"); second, the final score is determined by the relative ranking of the target paper's ideas. Additional evaluations include semantic similar-

*Table 3.* Details of Datasets in MemoryBench. In the "Task" column, "LiSo" indicates tasks with long inputs and short outputs, using 600 tokens as the threshold to differentiate between long and short. In the "Name" column, "A.K." refers to "Academic & Knowledge," "A.E." to "Academic & Engineering," "C.D." to "Creative & Design," and "P.L." to "Politics & Law." In the "Domain" column, "Open" denotes an Open-Domain. In the "Test Metrics" column, "Multi-Metrics" indicates that the dataset includes multiple evaluation metrics.

| Task | Name | Domain | Lang. | #Case | Test Metrics | Avg. Text Length Input | Output |
|---|---|---|---|---|---|---|---|
| LiSo | Locomo | Open | en | 1,986 | F1 | 22,426 | 10.11 |
| | DialSim-friends | Open | en | 1,000 | Accuracy | 334,490 | 2.98 |
| | DialSim-bigbang | Open | en | 1,000 | Accuracy | 365,678 | 3.77 |
| | DialSim-theoffice | Open | en | 1,000 | Accuracy | 383,054 | 4.29 |
| | LexEval-Summarization | Legal | zh | 1,000 | Rouge-L | 1,206 | 91.39 |
| | IdeaBench | Academic | en | 2,374 | Multi-Metrics | 1,077 | 322.78 |
| | LimitGen-Syn | Academic | en | 1,000 | LLM-as-Judge | 7,929 | 125.4 |
| SiLo | WritingPrompts | Open | en | 2,000 | METEOR | 34 | 692.48 |
| | JuDGE | Legal | zh | 2,505 | Multi-Metrics | 546 | 1,236.27 |
| | HelloBench-A.K.-QA | Academic | en | 213 | LLM-as-Judge | 76 | 1,628.04 |
| LiLo | HelloBench-C.D. | Open | en | 228 | LLM-as-Judge | 996 | 1,504.57 |
| | WritingBench-C.D. | Open | en,zh | 422 | LLM-as-Judge | 1,201 | 1,350.88 |
| | LexEval-Judge | Legal | zh | 1,000 | Rouge-L | 1,986 | 868.45 |
| | WritingBench-P.L. | Legal | en,zh | 201 | LLM-as-Judge | 2,363 | 1,493.96 |
| | HelloBench-A.K.-Writing | Academic | en | 82 | LLM-as-Judge | 1,437 | 1,179.24 |
| | WritingBench-A.E. | Academic | en,zh | 167 | LLM-as-Judge | 1,944 | 1,496.32 |
| SiSo | NF-Cats | Open | en | 2,397 | LLM-as-Judge | 50 | 188.34 |
| | LexEval-QA | Legal | zh | 500 | Rouge-L | 475 | 125.59 |
| | SciTechNews | Academic | en | 1,000 | Multi-Metrics | 277 | 160.52 |

ity (scored by BERTScore F1) and idea overlap (also scored by LLM-as-Judge) as reference metrics.

- **HelloBench** (Que et al., 2024): HelloBench evaluates the capability of LLM to generate long-form text across five tasks: open-ended question answering, chat, summarization, text completion, and heuristic text generation. Following the original paper's recommendation, we employ the "Average evaluation results of Checklists" method, which correlates most strongly with human judgments. In this approach, the LLM assigns various dimension scores based on a checklist for each question, and the average of these scores serves as the final result. From HelloBench, we derive three MemoryBench subsets:

  - **HelloBench-Academic&Knowledge-QA**: Comprises all open-ended QA tasks from HelloBench, as well as the "science problem solve" subset from the chat tasks.
  - **HelloBench-Academic&Knowledge-Writing**: Encompasses academic writing tasks, including the "academic article" subset in summarization, the "academic write" subset in chat, and the "argumentative writing" and "keyword writing" subsets in heuristic text generation.
  - **HelloBench-Creative&Design**: Focuses on creative and design tasks from HelloBench, encompassing "curriculum development," "character creation," "idea generation," "creative write," "script write," "continue write," and "guide generation" from the chat tasks; "roleplaying writing,"

"screenplay writing," and "story writing" from heuristic text generation; and all text completion tasks.

- **NF-Cats** (Bolotova et al., 2022): NF-Cats is a Non-Factoid Question-Answering dataset. We use its test set and follow the LLM as Judge evaluation approach recommended by NTCIR-18 (Chen et al., 2025). The prompt template for this evaluation is as follows:

> **Evaluation Prompt Template of NF-Cats for MemoryBench**
>
> ###Task: Evaluate the answer of a given question. Directly output an integer between 1 and 5 to indicate the score of this answer:
> - 1 means the answer is irrelevant to the question,
> - 2 means the answer is related to the question, but does not solve the question,
> - 3 means the answer only solves a part of the question,
> - 4 means the answer solve majority aspects of the question, but not perfect,
> - 5 means the answer is perfect to solve the question
>
> ###Question: { *Question* }
>
> ###Answer: { *Output* }

> ###Score of the answer:

### A.1.2. DETAILS OF METRIC INTEGRATION

Because certain datasets (JuDGE, IdeaBench, and SciTech-News) include multiple evaluation metrics, we employ LLM-as-Judge to merge these metrics into a single indicator. The corresponding prompt is as follows:

---

**Metric Integration Prompt Template for JuDGE**

You are an expert legal AI assistant. Your task is to evaluate the quality of an automatically generated legal judgment document based on the provided context and a set of pre-calculated metrics.

## Case Factual Description (Input)
{INPUT_FACTS}

## Generated Judgment Document (Output)
{GENERATED_JUDGMENT}

## Ground Truth Judgment Document (Reference)
{GOLDEN_JUDGMENT}

## Evaluation Metrics
Below are the calculated metrics comparing the 'Generated Judgment' to the 'Ground Truth'. A score of 1.00 indicates a perfect match for that specific metric, while 0.00 indicates a complete mismatch.

1. Penalty Accuracy (Scores range from 0.00 to 1.00)
time_score: {time_score} (Measures the accuracy of the prison sentence duration.)
amount_score: {amount_score} (Measures the accuracy of the monetary fine amount.)

2. Convicting Accuracy (Scores range from 0.00 to 1.00)
crime_recall: {crime_recall} (The proportion of actual charges that the system correctly identifies.)
crime_precision: {crime_precision} (The proportion of predicted charges that are accurate.)

3. Referencing Accuracy (Scores range from 0.00 to 1.00)
penalcode_index_recall: {penalcode_index_recall} (The proportion of correctly cited ground-truth statutes among all relevant statutes.)

penalcode_index_precision: {penalcode_index_precision} (The proportion of correctly cited statutes among all citations in the generated judgment.)
reasoning_meteor: {reasoning_meteor} (Semantic similarity of the 'Judicial Reasoning' section based on METEOR score.)
reasoning_bert_score: {reasoning_bert_score} (Semantic similarity of the 'Judicial Reasoning' section based on BERTScore.)
judge_meteor: {judge_meteor} (Semantic similarity of the 'Judgment Result' section based on METEOR score.)
judge_bert_score: {judge_bert_score} (Semantic similarity of the 'Judgment Result' section based on BERTScore.)

## Task
Based on a holistic review of the input, output, ground truth, and all the metrics provided above, provide a single integer score from 1 to 10 to represent the overall quality of the generated judgment document.
- 1: Represents extremely poor quality (e.g., completely irrelevant, factually incorrect, nonsensical).
- 10: Represents excellent quality (e.g., legally sound, factually accurate, well-reasoned, and structurally perfect, nearly indistinguishable from the ground truth).Your response should be only a single integer.

## Final Score

---

**Metric Integration Prompt Template for IdeaBench**

You are an expert scientific researcher and AI assistant. Your task is to evaluate the overall quality of an automatically generated research idea based on the provided context and a set of pre-calculated metrics.

## Background Knowledge (Input)
{INPUT_CONTEXT}

## Generated Research Idea (Output)
{GENERATED_IDEA}

## Ground Truth Research Idea (Reference)
{GOLDEN_IDEA}

## Evaluation Metrics

---

Below are the calculated metrics comparing the 'Generated Research Idea' to the 'Ground Truth'. Please use them to inform your overall score.

1. Semantic Similarity (bert_score): Measures the semantic similarity between the 'Generated Research Idea' and the 'Ground Truth Research Idea'. Scores range from 0.00 (no similarity) to 1.00 (perfect semantic match).
bert_score: {bert_score}

2. Idea Overlap (llm_rating_score): An LLM-based rating of the idea overlap between the 'Generated Research Idea' and the 'Ground Truth'. Scores range from 1 (minimal overlap) to 10 (perfect overlap).
llm_rating_score: {llm_rating_score}

3. Novelty Insight Score (llm_novelty_ranking_score): Quantifies the novelty of the 'Generated Research Idea' relative to the 'Ground Truth'. This score is derived by ranking the generated idea(s) against the ground truth idea. Scores range from 0.00 to 1.00.
* A score near **0.00** means the generated idea is significantly less novel than the ground truth.
* A score near **0.50** suggests comparable novelty.
* A score near **1.00** means the generated idea is significantly more novel than the ground truth.
llm_novelty_ranking_score:
{llm_novelty_ranking_score}

4. Feasibility Insight Score (llm_feasibility_ranking_score): Quantifies the feasibility of the 'Generated Research Idea' relative to the 'Ground Truth', using the same ranking methodology as the Novelty Insight Score. Scores range from 0.00 to 1.00.
* A score near **0.00** means the generated idea is significantly less feasible than the ground truth.
* A score near **0.50** suggests comparable feasibility.
* A score near **1.00** means the generated idea is significantly more feasible than the ground truth.
llm_feasibility_ranking_score:
{llm_feasibility_ranking_score}

## Task
Based on a holistic review of the input, output, ground truth, and all the metrics provided above, provide a single integer score from 1 to 10 to rep-

resent the overall quality of the generated research idea.
- 1: Represents extremely poor quality (e.g., incoherent, irrelevant, factually incorrect).
- 10: Represents excellent quality (e.g., coherent, insightful, novel, feasible, and well-aligned with the background knowledge, nearly indistinguishable from an idea proposed by a human expert).

Your response should be only a single integer.

## Final Score

Metric Integration Prompt Template for SciTech-News

You are an expert in science communication and text evaluation. Your task is to evaluate the quality of an automatically generated popular science article based on the provided source document, a reference article, and a set of pre-calculated metrics.

## Source Document (Input)
{INPUT_TEXT}

## Generated Popular Science Article (Output)
{GENERATED_ARTICLE}

## Abstract of Reference Popular Science Article (Golden Passage)
{GOLDEN_PASSAGE}

## Evaluation Metrics
Below are the calculated metrics comparing the 'Generated Article' to the 'Reference Article' or analyzing its intrinsic qualities.

Rouge-L (Score range: 0.00 to 1.00)
Score: {ROUGE_L}
Meaning: Measures the overlap of the longest common word sequence between the generated and reference articles. A higher score indicates better factual consistency and content preservation.

BERTScore-F1 (Score range: 0.00 to 1.00)
Score: {BERTSCORE_F1}
Meaning: Measures the semantic similarity between the generated and reference articles using contextual language models. A higher score indicates that the core meaning is better captured, even with different wording.

CLI (Coleman-Liau Index)
Score: {CLI}
Meaning: Estimates the U.S. grade level required to understand the text. For popular science, a lower score (e.g., 8-12) is generally desirable, indicating better readability and accessibility for a general audience.

FKGL (Flesch-Kincaid Grade Level)
Score: {FKGL}
Meaning: Similar to CLI, this metric also estimates the required U.S. grade level for comprehension. Lower scores suggest the text is easier to read. A score between 8 and 12 means standard readability for a general audience.

DCRS (Dale-Chall Readability Score)
Score: {DCRS}
Meaning: Estimates readability based on a list of 3000 common words. A lower score indicates the text is easier to understand. A score of 4.9 or lower indicates that the passage is very easy to read for fourth-grade students. A score between 9.0 and 9.9 indicates that the passage is at a college readability level.

## Task
Based on a holistic review of the input, output, golden passage, and all the metrics provided above, provide a single integer score from 1 to 10 to represent the overall quality of the generated popular science article. Consider its accuracy, readability, coherence, and faithfulness to the source material.
- 1: Represents extremely poor quality (e.g., completely irrelevant, factually incorrect, nonsensical, or unreadable).
- 10: Represents excellent quality (e.g., accurate, easy to understand for a layperson, well-structured, engaging, and highly faithful to the source, nearly indistinguishable from the reference).

Your response should be only a single integer.

## Final Score

## A.2. Feedback Simulation Details

This appendix provides a detailed description of our user feedback simulation framework, which is designed to generate realistic and multifaceted feedback signals for evaluating the continual learning capabilities of Large Language Model systems (LLMsys). We detail the framework's architecture, the implementation of the LLM-as-User-Simulator, the probabilistic model for user actions, and the specific prompt templates used in our experiments.

### A.2.1. MULTI-PATH SIMULATION ARCHITECTURE

The simulation process begins by identifying the dataset type to determine the appropriate evaluation path. This dual-path architecture ensures both high-fidelity evaluation for complex, subjective tasks and computational efficiency for tasks with objective success criteria. The ultimate output for any given interaction is a comprehensive tuple containing a session termination signal, a natural language user response, a simulated user interface action, and a numerical satisfaction score.

**Path 1: Metric-Based Direct Evaluation** This path is reserved for tasks where an objective, computable metric can definitively assess the correctness of an LLMsys's response. In our work, this primarily applies to reading comprehension tasks (e.g., "needle-in-a-haystack") from datasets like Locomo and DialSim, where the LLMsys must retrieve a specific piece of information from a large corpus.

For efficiency and to avoid exposing extensive context to the simulator LLM, we use a streamlined approach that directly converts objective metrics into satisfaction scores:

- **DialSim:** We use boolean accuracy to determine correctness, then assign deterministic satisfaction scores: $S = 3$ for incorrect responses and $S = 9$ for correct responses. This binary scoring reflects the objective nature of the task while maintaining compatibility with our probabilistic action model.

- **Locomo:** We compute the F1 score between the LLMsys's extracted answer and the ground truth, then map it deterministically to the 1-10 satisfaction scale. For example: F1 $\geq 0.9$ yields $S = 10$, F1 $\geq 0.8$ yields $S = 9$, F1 $\geq 0.5$ (correctness threshold) yields $S = 6$, with lower F1 scores mapped to correspondingly lower satisfaction scores.

This design choice prioritizes computational efficiency while maintaining evaluation quality. Both datasets use deterministic scoring that eliminates the need for costly LLM-based evaluation: DialSim's binary mapping preserves the essential correctness signal, while Locomo's F1-based mapping captures nuanced quality variations that reflect partial correctness in retrieval tasks.

**Path 2: LLM-as-User-Simulator** For the majority of datasets involving more nuanced tasks like writing, coding, or open-ended question answering, a simple metric is insufficient. For these, we employ an LLM-as-User-Simulator. We leverage a strong LLM (e.g., `Qwen-3-32B`) to act as

a proxy for a human user, providing rich, contextual feedback. The core of this mechanism lies in a meticulously engineered prompting strategy, detailed below.

### A.2.2. SIMULATION IMPLEMENTATION

The simulator's ability to generate human-like feedback is contingent on a structured and comprehensive prompting system. This system is composed of two distinct prompts: a profile prompt to define the user's persona and a test prompt to provide the context for a specific evaluation.

**Prompting Strategy**

**Profile Prompt Construction:** The profile prompt (Prompt A.2.4.1) establishes the stable characteristics and behavioral rules for the simulated user. It is composed of four key elements:

- **User Persona:** A description of the simulated user's identity and general disposition (e.g., "You are a software engineer reviewing a code snippet.").

- **Domain Expertise:** A specification of the user's knowledge level relevant to the task.

- **Evaluation Criteria:** A list of dataset-specific dimensions for judging the LLMsys's response (e.g., clarity, coherence, and creativity for a writing task).

- **Behavioral Constraints:** A set of fixed rules, such as focusing on the user's initial request and adhering to the specified output format.

**Test Prompt Construction:** The test prompt (Prompt A.2.4.2) provides the immediate context for evaluating a single conversational turn. It aggregates all necessary information:

- **Conversation History:** The full history of the current interaction.

- **Task Description:** A concise summary of the end-user's underlying goal.

- **Evaluation Context:** Ground truth information (e.g., a golden answer or reference solution) that allows the simulator to act as an oracle.

- **Language Constraint:** An instruction for the simulator to respond in the same language as the conversation.

- **JSON Output Format:** A strict instruction to format the entire output as a machine-readable JSON object.

**Feedback Generation Process** The simulator generates a structured JSON object containing multiple facets of feedback. This process unfolds in two main stages:

1. **Qualitative Feedback Generation:** The primary LLM-as-User-Simulator is prompted to produce a JSON object containing two key fields. The first, reasoning, provides a detailed, chain-of-thought analysis of the LLMsys's response quality. The second, response, contains the natural language utterance the user would say next (if continuing the conversation).

2. **Quantitative Satisfaction Scoring:** For datasets using Path 1 (DialSim and Locomo), satisfaction scores are generated deterministically as described above. For Path 2 datasets, a separate LLM-based module, the *SatisfactionScorer*, evaluates the LLMsys's last response using ground truth context and assigns a numerical satisfaction score $S$ on a 1-10 scale, guided by Prompts A.2.4.3 and A.2.4.4. The resulting score quantifies overall response quality and serves as the primary input for modeling subsequent user actions.

### A.2.3. PROBABILISTIC USER ACTION MODELING

To translate the numerical satisfaction scores into realistic user actions (like, dislike, copy, or no action), we developed a probabilistic feedback model. The model implementation uses three distinct approaches to accommodate fundamentally different scoring characteristics: (1) a binary model for DialSim's two-score system, (2) a deterministic sigmoid model for Locomo's F1-based scoring, and (3) a general sigmoid model for all other LLM-scored datasets. We calculate separate action probability mappings for each approach while maintaining the same target global probabilities across all datasets.

We model the conditional probabilities of a user liking ($L$), disliking ($D$), or taking no action ($N$) given a score $S$. The core assumption is that the propensity to like is monotonically increasing with satisfaction, while the propensity to dislike is monotonically decreasing. We use scaled logistic (sigmoid) functions to capture this behavior:

$$P(L|S) = c_L \cdot \sigma(k_L(S - S_{0L})) \tag{1}$$

$$P(D|S) = c_D \cdot \sigma(-k_D(S - S_{0D})) \tag{2}$$

where $\sigma(x) = (1 + e^{-x})^{-1}$ is the sigmoid function. The parameters $(k_L, S_{0L})$ and $(k_D, S_{0D})$ are hyperparameters that control the steepness and midpoint of the like/dislike probability curves, respectively.

**General Sigmoid Model (LLM-Scored Datasets)** For the majority of datasets using LLM-based satisfaction scoring that naturally produces the full 1-10 score range, we

*Table 4.* The empirical distribution of LLM-generated user satisfaction scores ($S$).

| Score ($S$) | 1 | 2 | 3 | 4 | 5 | 6 | 7 | 8 | 9 | 10 |
|---|---|---|---|---|---|---|---|---|---|---|
| **Percentage** | 0.02% | 0.93% | 3.06% | 1.93% | 0.40% | 2.56% | 17.5% | 32.12% | 41.05% | 0.43% |

model the conditional probabilities using scaled logistic (sigmoid) functions as defined in Equations 1 and 2. The model's macro-level parameters are designed to be tunable for different application contexts.

For the experiments in this paper, we instantiate these parameters by adopting statistical findings from large-scale user studies. Specifically, Shuster et al. (2022) report that about $6.5\%$ of users provide explicit feedback in the form of actions, and Chatterji et al. (2025) further show that such feedback exhibits an approximately $86\%$ to $14\%$ like-to-dislike ratio. Following these empirical findings, we set the overall user feedback rate to $6.5\%$ with the same like-to-dislike split. This yields target global probabilities of $P(L) \approx 0.0559$ and $P(D) \approx 0.0091$.

Based on these constraints and our score distribution, we set hyperparameters to $S_{0L} = 7.5$, $S_{0D} = 4.5$, and $k_L = k_D = 1.5$, ensuring low scores predominantly trigger dislikes and high scores trigger likes. The scaling constants $c_L$ and $c_D$ are analytically derived to match target global probabilities.

**Binary Model (DialSim)** For DialSim's binary scoring system, we use a simplified probabilistic mapping that directly assigns action probabilities to the two possible scores:

- For incorrect responses ($S = 3$): $P(D|S = 3) = \frac{P(D)}{P(S=3)}$ and $P(L|S = 3) = 0$

- For correct responses ($S = 9$): $P(L|S = 9) = \frac{P(L)}{P(S=9)}$ and $P(D|S = 9) = 0$

This approach maintains the same target global probabilities while accommodating the binary nature of DialSim's evaluation, resulting in $P(L|S = 9) \approx 9.9\%$ and $P(D|S = 3) \approx 2.1\%$ based on the empirical score distribution.

**Deterministic Sigmoid Model (Locomo)** Locomo datasets require separate treatment due to their deterministic F1-based satisfaction scoring mechanism. Although F1 scores are continuous and map to the full 1-10 range, the deterministic nature of this mapping creates a fundamentally different score distribution compared to LLM-generated scores. We therefore apply the same sigmoid model formulation as other datasets but compute separate scaling constants $(c_L, c_D)$ specifically calibrated to Locomo's empirical score distribution. This ensures appropriate action

probabilities while maintaining the target global rates. Since Locomo's distribution pattern differs significantly from the general case, separate probability mappings are essential for realistic user behavior simulation.

**Copy Action Generation** Beyond like and dislike actions, users may also copy helpful responses for future reference. Copy behavior is task-dependent: for creative or long-form generation tasks (SiLo and LiLo task types), users are more likely to save comprehensive responses, while for short-answer tasks (LiSo and SiSo), users typically just read the results without copying. We model copy probability as four times the like probability for long-output tasks and zero for short-ouput tasks:

$$P(C|S) = \begin{cases} 4P(L|S) & \text{if task generates long output} \\ 0 & \text{if task generates short output} \end{cases}$$
(3)

Importantly, the hyperparameters $(k, S_0)$ are tunable. Future work building on our system can adjust these values to simulate different user populations, such as those who are more or less critical. The resulting conditional probabilities for the general sigmoid model (representing LLM-scored datasets) are presented in Table 5. DialSim and Locomo datasets use their respective separate probability mappings as described above.

A.2.4. PROMPT TEMPLATES

This section contains the specific prompt templates used for the LLM-as-User-Simulator and the SatisfactionScorer modules. We summarize the general-purpose templates first (Prompts A.2.4.1–A.2.4.4), followed by the SciTechNews specialization (Prompts A.2.4.5–A.2.4.7).

---

**Overall Profile Prompt**

{user_persona}

{domain_expertise}

CRITICAL: Always focus on the initial prompt/request as the primary context for evaluation. The conversation should stay aligned with the original user intent.

IMPORTANT: DO NOT REPEAT QUESTIONS

---

*Table 5.* Final conditional probabilities of user actions based on satisfaction score (general sigmoid model for LLM-scored datasets).

| Score ($S$) | Like Prob. $P(L|S)$ | Dislike Prob. $P(D|S)$ | No Action Prob. $P(N|S)$ |
|---|---|---|---|
| 1 | 0.000% | 15.091% | 84.909% |
| 2 | 0.002% | 14.821% | 85.177% |
| 3 | 0.010% | 13.723% | 86.267% |
| 4 | 0.045% | 10.303% | 89.652% |
| 5 | 0.197% | 4.867% | 94.936% |
| 6 | 0.817% | 1.446% | 97.737% |
| 7 | 2.748% | 0.349% | 96.903% |
| 8 | 5.818% | 0.079% | 94.103% |
| 9 | 7.749% | 0.018% | 92.233% |
| 10 | 8.369% | 0.004% | 91.627% |

OR REQUESTS that have already been asked in the conversation. Avoid asking the same question multiple times.

IMPORTANT: Always start your reasoning process first, then provide the other feedback elements.

Your response should include:
1. Reasoning: Detailed analysis of the assistant's response quality and accuracy (always consider how well it addresses the initial prompt)
2. Behavior decision: Whether to continue or end the conversation
3. Response: What the user would say (only if continuing the conversation)

Consider factors like: {evaluation_criteria}

{additional_context}

### Overall Test Prompt

Analyze this conversation and predict the user's response:

The user is {task_description}. CRITICAL: Focus on the initial request as the core topic that should be the primary focus throughout this entire conversation. All responses should be evaluated based on how well they address this original user intent.

Conversation History: {conversation_history}

EVALUATION CONTEXT: {evaluation_context}

IMPORTANT: If you provide a response (when

behavior is continue_conversation), it must be in {language}.

Please provide a realistic user response in strict JSON format:
{
    "reasoning": "Detailed analysis of the assistant's response quality and accuracy (MUST evaluate how well it addresses the initial request)",
    "behavior": "continue_conversation" — "end_conversation",
    "response": "What the user would say next (string or null if ending)"
}

Requirements:
- reasoning: Always provide detailed analysis first. CRITICAL: Always assess how well the assistant's response addresses the initial request and stays focused on the original user intent.
- behavior: Must be exactly: continue_conversation or end_conversation.
- response: Text if continuing, null if ending conversation. Must match the conversation language. IMPORTANT: Do not repeat questions or requests that have already been made in the conversation.

Respond with valid JSON only.

### Satisfaction Scoring System Prompt

You are an expert evaluator tasked with scoring assistant responses against specific quality standards.

SCORING SCALE (1-10):
1-2: Completely inadequate - Wrong, irrelevant, or harmful

3-4: Unsatisfactory - Major errors, misses key points, or unhelpful

5-6: Below expectations - Addresses basics but has significant gaps, inaccuracies, or omissions

7-8: Meets expectations - Solid response with minor issues or missing elements

9-10: Exceeds expectations - Comprehensive, accurate, and fully satisfies all requirements

EVALUATION APPROACH:
- Use the provided evaluation context and ground truth as your primary standards
- Score against what the response should contain, not just what it does contain
- Consider both correctness and completeness

Provide only a numerical score from 1-10.

---

**Satisfaction Scoring User Prompt**

Evaluate the assistant's response by comparing it against the provided standards and ground truth:

FULL CONVERSATION: {conversation_history}

EVALUATION CONTEXT (contains ground truth and quality criteria): {evaluation_context}

EVALUATION TASK: Compare the assistant's final response against the evaluation context above. The evaluation context contains the ground truth and quality standards that define what a good response should include.

Respond in this JSON format:
```
{
    "score": <integer from 1-10>
}
```

---

**Example: SciTechNews Dataset Prompt**

Below is a concrete example of the prompts used for the SciTechNews dataset, which focuses on science journalism evaluation:

---

**SciTechNews Profile Prompt**

You are simulating a science journalist or editor who requested AI assistance to write journalistic reports of scientific papers for general audiences.

You have expertise in science journalism across diverse fields including computer science, cyberse-

curity, privacy research, mobile computing, cloud services, encryption technologies, biomedical research, environmental science, and other technical domains. You understand what makes scientific writing accessible to the general public while maintaining accuracy.

CRITICAL: Always focus on the initial prompt/request as the primary context for evaluation. The conversation should stay aligned with the original user intent.

IMPORTANT: DO NOT REPEAT QUESTIONS OR REQUESTS that have already been asked in the conversation. Avoid asking the same question multiple times.

IMPORTANT: Always start your reasoning process first, then provide the other feedback elements.

Your response should include:
1. Reasoning: Detailed analysis of the assistant's response quality and accuracy (always consider how well it addresses the initial prompt)
2. Behavior decision: Whether to continue or end the conversation
3. Response: What the user would say (only if continuing the conversation)

Consider factors like:
- Accessible and readable for general audiences without technical background
- Accurate to the original scientific work without oversimplification
- Engaging and newsworthy in its presentation style
- Well-structured with appropriate journalistic elements (headlines, lead paragraphs, context)
- Properly balancing technical detail with readability
- Readability for lay audiences
- Journalistic style and structure
- Engagement factor and clarity of technical concepts

Your evaluation focuses on the journalistic transformation of academic content rather than the underlying research quality. Consider: readability for lay audiences, accuracy to source material, journalistic style and structure, engagement factor, and clarity of technical concepts.

---

**SciTechNews Test Prompt**

Analyze this conversation and predict the user's response:

The user is seeking to transform scientific papers into accessible journalistic reports for general audiences. CRITICAL: Focus on the initial request as the core topic that should be the primary focus throughout this entire conversation. All responses should be evaluated based on how well they address this original user intent.

Conversation History: {conversation_history}

EVALUATION CONTEXT:
Reference Popular Title: {pr_title}
Reference Popular Abstract: {pr_abstract}
Use this reference to evaluate the quality of the journalistic transformation. Consider how well the generated content compares to this reference in terms of accessibility, accuracy, and engagement for general audiences.

IMPORTANT: If you provide a response (when behavior is continue_conversation), it must be in English.

Please provide a realistic user response in strict JSON format:
{
    "reasoning": "Detailed analysis of the assistant's response quality and accuracy (MUST evaluate how well it addresses the initial request)",
    "behavior": "continue_conversation" —
"end_conversation",
    "response": "What the user would say next (string or null if ending)"
}

Requirements:
- reasoning: Always provide detailed analysis first. CRITICAL: Always assess how well the assistant's response addresses the initial request and stays focused on the original user intent.
- behavior: Must be exactly: continue_conversation or end_conversation
- response: Text if continuing, null if ending conversation. Must match the conversation language. IMPORTANT: Do not repeat questions or requests that have already been made in the conversation.

Respond with valid JSON only.

---

**SciTechNews Satisfaction Scoring User Prompt**

Evaluate the assistant's response by comparing it against the provided standards and ground truth:

FULL CONVERSATION: {conversation_history}

EVALUATION CONTEXT (contains ground truth and quality criteria):
Reference Popular Title: {pr_title}
Reference Popular Abstract: {pr_abstract}
Evaluation Criteria:
- Accessible and readable for general audiences
- Accurate to original scientific work
- Engaging and newsworthy presentation
- Well-structured journalistic elements

EVALUATION TASK: Compare the assistant's final response against the evaluation context above. The evaluation context contains the ground truth and quality standards that define what a good response should include.

Respond in this JSON format:
{
    "score": <integer from 1-10>
}

---

### A.2.5. HUMAN ANNOTATION OF FEEDBACK GENERATION QUALITY

To better evaluate the quality of simulated user feedback in MemoryBench, we randomly sampled 200 tasks across 4 task format categories, 3 domains, and 2 languages together with feedback generated by the LLM-as-User-Simulator from the training set in the off-policy experimental setting for human annotation. The distribution of the sampled task cases is shown in Table 6. For each task case, we conducted a two-step human annotation. First, we hired professional human annotators to write verbal feedback for each task based on the task needs and evaluation criteria. Second, we conducted crowdsourcing by recruiting 9 annotators (mostly undergraduate and graduate students on campus) to conduct pairwise annotations to compare the LLM-simulated user feedback with human feedback in terms of language naturalness, relevance to the task context, and which looks more like a feedback written by human users:

- **Naturalness**: Whether the feedback sounds like something a real user would naturally say, using authentic vocabulary, tone, and flow without unnatural repetition or stiffness.

- **Relevance**: Whether the feedback stays on-topic, be

logically coherent, and provide specific, valuable guidance or information related to the conversation.

- **Overall Quality**: Considering both naturalness and relevance, whether the feedback looks like a plausible, reasonable response from a real user in the given context.

Each task case is annotated by three different annotators. Ideally, **if the feedback generated by our framework is indistinguishable from those written by human, either the annotators should find the simulated feedback is more human-like than the human written ones, or the annotators' agreements (e.g., Fleiss' Kappa) on which feedback is more human-like should be low**. The annotation results are shown in Table 7.

As shown in the Table 7, the Kappa value of Naturalness is low, which means that it was difficult for the annotators to consistently distinguish whether the human-written or LLM-simulated user feedback is more natural in language. This indicates that, in general, it is difficult to distinguish human-written verbal feedback with simulated verbal feedback generated by the LLM-as-user framework in MemoryBench. In Table 7, half of the annotators believe that the simulated feedback is more natural than human-written feedback, and a slightly more than half of the annotators feel that the simulated feedback is more likely to be human-written than the actual human-written feedback from the perspective of relevance and overall quality. These are supportive evidences showing that our LLM-as-user framework can generate high-quality user feedback that are indistinguishable from real human feedback.

## A.3. Additional Experiment Details and Results

### A.3.1. DATASETS MIXTURES DETAILS

For each dataset category, we randomly sampled up to 250 data points with a fixed random seed (if the dataset contained fewer than 250 samples, we used all available samples). The sampled data were split into training and test sets with a 4:1 ratio, and then merged across datasets to construct the final training and test sets for each category. Table 8 summarizes the datasets included in each domain, along with their total sizes and the number of sampled instances used in our experiments. Table 9 provides a similar overview for each task , showing how datasets are distributed and sampled within each category. We also provide an extended version of MemoryBench using all data from all datasets with the same training and test sets sampling ratio as MemoryBench-Full[5]. However, the sizes of different datasets in MemoryBench-Full vary significantly, which may require special handling

---

[5]https://huggingface.co/datasets/THUIR/MemoryBench-Full

when calculating averages to prevent any single dataset from disproportionately influencing the results, so we do not include the experiments on it in this paper.

### A.3.2. BASELINES

We compare 8 LLMsys, including direct generation without memory, naive RAG systems, and several newly proposed memory systems. The baselines are as follows:

- **Vanilla**: A simple baseline without any memory mechanism, where the LLM directly generates a response to the query of each data point.

- **BM25-M**: A RAG system that uses BM25 as the retriever. For a dialogue, each message is stored as an individual retrieval entry, together with the dialogue ID and its position within the dialogue.

- **BM25-S**: Another BM25-based RAG system. Unlike BM25-M, here the entire dialogue is stored as a single retrieval entry, indexed with the question that generated the dialogue.

- **Embed-M**: A RAG system using the embedding model Qwen3-Embedding-0.6B for vector-based retrieval. For a dialogue, each message is encoded and stored as a separate retrieval entry, together with the dialogue ID and its position within the dialogue.

- **Embed-S**: Similar to Embed-M, but each full dialogue is encoded and stored as a single retrieval entry, indexed by the corresponding question.

- **A-Mem** (Xu et al., 2025a): An agentic memory system for LLM agents, designed to dynamically organize and evolve memories rather than relying on fixed structures. A-Mem consists of three major modules: memory note construction, dynamic memory linking for continual evolution, and memory retrieval. Specifically, A-Mem uses an LLM to extract a structured memory note for each input text and encodes these notes into embeddings. It then uses embedding similarity to retrieve the most relevant existing memories, constructs memory links between them, and updates linked memories jointly via LLM prompting. During inference, the most relevant memory notes are retrieved and subsequently incorporated into the context to augment the model's generation. We adopt its official implementation with the provided APIs for adding and deleting memories, and use the default embedding model all-MiniLM-L6-v2. Following its implementation on the LoCoMo dataset, each message of a dialogue is stored as an independent memory entry, along with the dialogue ID and the message position.

*Table 6.* Distribution of sampled task cases for human annotation.

| Divided by | Language | | Task | | | | Domain | | |
|---|---|---|---|---|---|---|---|---|---|
| | **English** | **Chinese** | **LiSo** | **SiLo** | **LiLo** | **SiSo** | **Open** | **Academic** | **Legal** |
| **Samples** | 131 | 69 | 43 | 46 | 74 | 37 | 66 | 77 | 57 |

*Table 7.* Human annotation results comparing the simulated and human-written feedback. If the simulated feedback is indistinguishable from human feedback, the Simulated Feedback Win-rate should be higher than 50% or the annotators' agreements on which feedback is more human-like (i.e., Fleiss' Kappa) should be low.

| | Annotator Kappa | Simulated Feedback Win-rate | Human Feedback Win-rate |
|---|---|---|---|
| Naturalness | 0.093 | 50.0% | 50.0% |
| Relevance | 0.380 | 59.5% | 40.5% |
| Overall Quality | 0.273 | 59.0% | 41.0% |

- **Mem0** (Chhikara et al., 2025): A scalable memory-centric architecture that dynamically extracts, consolidates, and retrieves salient information from ongoing conversations. Mem0 includes two core modules: memory extraction and memory update. For extraction, Mem0 uses dialogue summaries and recent messages to generate candidate memories. For update, Mem0 leverages the LLM's inherent reasoning capability for efficient memory maintenance. The LLM compares new information against existing memories and chooses among four explicit operations—ADD, UPDATE, DELETE, or NOOP—to guarantee data consistency and minimize redundancy in the memory store. Mem0 also proposes an extension, Mem0$^g$, which structures memories as a knowledge graph; however, Mem0$^g$ is not used in our experiments. We use its open-source version with the provided APIs for memory addition and deletion, and employ Qwen3-Embedding-0.6B as the embedding model. According to the input forms of the API for memory addition, each full dialogue is added as a single memory entry. All memories, including both declarative memories from the dataset and procedural memories generated during interactions, are attributed to the same user identity.

- **MemoryOS** (Kang et al., 2025): A memory operating system that introduces a hierarchical and comprehensive memory management framework for LLM agents, inspired by operating system principles. MemoryOS includes four core modules: memory storage, update, retrieval, and generation. Analogous to an operating system, it organizes memories into short-term, mid-term, and long-term storage units, and refreshes them through a segmented paging architecture driven by dialogue-chain dynamics and heat-based mechanisms. During retrieval, MemoryOS performs semantic segmentation to query memories across different layers, and integrates the retrieved segments into a coherent prompt that supports downstream generation. We adopt its official implementation with the provided

API for adding memories and use all-MiniLM-L6-v2 as the embedding model. As required by its API, we combine one user message with its corresponding assistant reply into a single memory entry. All memories are attributed to the same user identity. In particular, for compatibility with Chinese datasets, we translated several internal English prompts into Chinese when used in Chinese datasets.

For A-Mem, Mem0, and MemoryOS, we directly use their released code with default parameter settings.

We use the same backbone LLM (i.e. Qwen3-8B) for all LLMsys, set the generation temperature to 0.1, top_p to 0.1, top_k to 1, and restrict the maximum generation length to 2,048 tokens.

Among all baselines, only the four RAG-based systems, A-Mem, and Mem0 allow to configure the number of retrieved memory entries. For these systems, we set the default retrieval number to five entries per query. If the resulting input exceeds the context length limit of the backbone LLM, the number of retrieved entries is gradually reduced until generation is feasible.

For a query and its retrieved memory entries, the input prompt for LLMsys is constructed in the following format:

---
Prompt Template for answering questions in English

User Memories:
{memories_str}

User input:
{query}

Based on the memories provided, respond naturally and appropriately to the user's input above.

---

*Table 8.* Datasets used in each domain category. "Total" denotes the full size of each dataset, while "Samples" indicates the number of instances selected for our experiments.

| Domain | Dataset Name | Total | Samples |
|---|---|---|---|
| Open | LoCoMo | 1986 | 250 |
| | DialSim-friends | 300 | 83 |
| | DialSim-bigbang | 300 | 83 |
| | DialSim-theoffice | 300 | 84 |
| | HelloBench-C.D. | 228 | 228 |
| | WritingPrompts | 2000 | 250 |
| | WritingBench-C.D. | 422 | 250 |
| | NF-Cats | 2397 | 250 |
| | **Total** | **10037** | **1478** |
| Legal | JuDGE | 2505 | 250 |
| | LexEval-Summarization | 1000 | 250 |
| | LexEval-Judge | 1000 | 250 |
| | LexEval-QA | 500 | 250 |
| | WritingBench-P.L. | 201 | 201 |
| | **Total** | **5206** | **1201** |
| Academic | HelloBench-A.K.-QA | 213 | 213 |
| | HelloBench-A.K.-Writing | 82 | 82 |
| | IdeaBench | 2374 | 250 |
| | SciTechNews | 1000 | 250 |
| | LimitGen-Syn | 1000 | 250 |
| | WritingBench-A.E. | 167 | 167 |
| | **Total** | **4836** | **1212** |

### A.3.3. EXPERIMENT SETTINGS

We design both off-policy and on-policy experiments to comprehensively evaluate the performance of LLMsys.

In the off-policy experiments, we begin by generate dialogues based on the training data to facilitate LLMsys learning. For datasets without declarative memory, the backbone LLM interacts with the User Feedback Simulator for up to three turns per question. The first turn consists of the question and the LLM's initial response, while the subsequent two turns involve iterative interactions between the simulator and the LLMsys, allowing the system to refine its outputs based on simulated feedback.

For datasets containing declarative memory (e.g., LoCoMo and DialSim), only LLMsys equipped with memory mechanisms can effectively process these cases. For each question, the LLMsys first loads the relevant knowledge into its memory before engaging in up to three turns of interaction with the User Feedback Simulator. In this process, only the first turn queries the memory; the following turns are based on the initially retrieved information. Since different LLMsys manage and utilize memory in distinct ways, we generate separate logs for each system accordingly. The demo of the pre-generated dialogues for the datasets are available in

here[6].

Once the training dialogues are generated, we evaluate the ability of LLMsys to answer questions in the test sets. For the Vanilla baseline, responses are generated directly without relying on any memory. For memory-augmented systems, the previously generated training dialogues are first loaded into the memory before generating responses for test set questions. By default, these memory systems only store the training dialogues. For datasets containing static knowledge, the memory systems include both all training dialogues and the static knowledge relevant to the current question, ensuring that the system has sufficient context to generate accurate answers.

Evaluation is conducted at the level of individual questions using the original metrics specified by each dataset. The results are then normalized and aggregated across datasets to produce overall performance scores.

A special consideration arises for datasets that contain static knowledge, such as LoCoMo and DialSim. These datasets have some conversation sessions as their static knowledge

---

[6]https://anonymous.4open.science/r/
MemoryBench-Dataset-BF1F

*Table 9.* Datasets used in each task category. "Total" denotes the full size of each dataset, while "Samples" indicates the number of instances selected for our experiments.

| Task | Dataset Name | Total | Samples |
|------|-------------|-------|---------|
| LiSo | LoCoMo | 1986 | 250 |
| | DialSim-friends | 300 | 83 |
| | DialSim-bigbang | 300 | 83 |
| | DialSim-theoffice | 300 | 84 |
| | LexEval-Summarization | 1000 | 250 |
| | IdeaBench | 2374 | 250 |
| | LimitGen-Syn | 1000 | 250 |
| | **Total** | **9364** | **1250** |
| SiLo | JuDGE | 2505 | 250 |
| | HelloBench-A.K.-QA | 213 | 213 |
| | WritingPrompts | 2000 | 250 |
| | **Total** | **4718** | **713** |
| LiLo | LexEval-Judge | **1000** | **250** |
| | WritingBench-P.L. | 201 | 201 |
| | HelloBench-A.K.-Writing | 82 | 82 |
| | WritingBench-A.E. | 167 | 167 |
| | HelloBench-C.D. | 228 | 228 |
| | WritingBench-C.D. | 422 | 250 |
| | **Total** | **2100** | **1178** |
| SiSo | LexEval-QA | 500 | 250 |
| | SciTechNews | 1000 | 250 |
| | NF-Cats | 2397 | 250 |
| | **Total** | **1500** | **750** |

and their questions cannot be answered correctly without access to the relevant knowledge. For the Vanilla baseline, we appended the first $n$ conversation sessions (where $n$ is the maximum number of sessions that fit within the model's context window) directly to the input context to support reasoning. In contrast, Mem0 was unable to complete memorizing the entire static knowledge corpus in DialSim-theoffice within a reasonable amount of time (see detailed explanation in Appendix A.3.8). Therefore, for all tasks involving this dataset, Mem0 was excluded from evaluation.

### A.3.4. OFF-POLICY EXPERIMENTS

We constructed multiple LLMsys using Qwen3-8B as the backbone model and evaluated their performance on MemoryBench. To generate the training dialogues used in the off-policy setting, we employed Qwen3-32B as the feedback simulator model and Qwen3-8B as the assistant model. In the off-policy experiments, the LLMsys are required to leverage the dialogues generated on the training set in order to better solve problems in the test set. For each domain and task, we aggregate the results by reporting both the min-max normalization score and the Z-score. The results

of min-max normalization scores and z-score scores are shown in Table 10 and Table 11 respectively. To enable a cross-experimental comparison, the min and max values used for min-max normalization in all other experiments are aligned with those obtained from the Qwen3-8B off-policy experiments on each dataset.

To provide a complete view, we also present the detailed evaluation results before aggregation in Table 12 and the variance in Table 13&14.

In addition to Qwen3-8B, we also explored building LLMsys using a larger backbone model, Qwen3-32B. To ensure consistency, we used the same training dialogues as procedural memory that were generated with Qwen3-8B, which are also released as part of our public dataset. The experimental results with Qwen3-32B are reported in Table 15.

### A.3.5. FEEDBACK EFFECT EVALUATION

To evaluate the quality of feedback generated by our user feedback simulator, we compare the performance with and without feedback. Specifically, for each data point, we let LLMsys and the feedback simulator interact for up to three

*Table 10.* Off-policy experimental results with Qwen3-8B as the backbone model. The evaluation metric is the average min-max normalization scores of each dataset.

| LLMsys | Domain | | | Task | | | |
|---|---|---|---|---|---|---|---|
| | Open | Academic | Legal | LiSo | SiLo | LiLo | SiSo |
| **Vanilla** | 0.6523 | 0.7154 | 0.4637 | 0.4091 | **0.7517** | **0.6417** | 0.7302 |
| **BM25-M** | 0.5720 | 0.6931 | **0.5011** | 0.3869 | 0.7287 | 0.6318 | 0.7275 |
| **BM25-S** | 0.6503 | 0.7110 | 0.4776 | 0.4560 | 0.7260 | 0.6310 | 0.7330 |
| **Embed-M** | 0.5760 | 0.7051 | 0.4635 | 0.3844 | 0.7120 | 0.5908 | 0.6919 |
| **Embed-S** | **0.6582** | **0.7185** | 0.4806 | **0.4661** | 0.7325 | 0.6328 | 0.7305 |
| **A-Mem** | 0.6243 | 0.7106 | 0.4684 | 0.4303 | 0.7172 | 0.6139 | 0.7147 |
| **Mem0** | - | 0.6965 | 0.4451 | - | 0.7305 | 0.5957 | 0.6745 |
| **MemoryOS** | 0.5894 | 0.6212 | 0.4290 | 0.3711 | 0.7493 | 0.4879 | **0.7452** |

*Table 11.* Off-policy experimental results with Qwen3-8B as the backbone model. The evaluation metric is Z-score.

| LLMsys | Domain | | | Task | | | |
|---|---|---|---|---|---|---|---|
| | Open | Academic | Legal | LiSo | SiLo | LiLo | SiSo |
| **Vanilla** | **0.1583** | 0.1450 | -0.0019 | 0.0162 | **-0.0467** | **0.1828** | 0.0392 |
| **BM25-M** | -0.1813 | 0.0293 | **0.1531** | -0.0375 | -0.1930 | 0.1166 | -0.0641 |
| **BM25-S** | 0.1030 | 0.1265 | 0.0496 | 0.1201 | -0.2169 | 0.1339 | 0.0855 |
| **Embed-M** | -0.1579 | 0.0760 | 0.0081 | -0.0514 | -0.2970 | -0.0343 | -0.1608 |
| **Embed-S** | 0.1081 | **0.1775** | 0.0669 | **0.1629** | -0.1718 | 0.1373 | 0.0495 |
| **A-Mem** | 0.0340 | 0.1124 | 0.0458 | 0.0743 | -0.2650 | 0.0555 | -0.0814 |
| **Mem0** | - | 0.0045 | -0.0843 | - | -0.1836 | -0.0049 | -0.2593 |
| **MemoryOS** | -0.0875 | -0.2621 | -0.1561 | -0.0824 | -0.0619 | -0.4876 | **0.1074** |

turns. The first response of LLMsys to the current question is regarded as the without-feedback output (consistent with the Vanilla baseline). We then feed the entire dialogue history back into the LLMsys and let it answer the same question again; this response is treated as the with-feedback output.

Firstly, we conduct this comparison across all datasets that do not involve external knowledge. For datasets that does not evaluated by LLM-as-judge, we use the full dataset for comparison. For datasets evaluated by LLM-as-judge (including HelloBench, IdeaBench, LimitGen-Syn, and NF-Cats), we sample the first 100 questions for comparison. The results are shown in Table 16. We find that on most datasets, LLMsys achieves higher scores when feedback is provided, suggesting that feedback helps the system improve its answers to the current question.

Moreover, although datasets with external knowledge (including LoCoMo and DialSim) only use fixed, non-LLM-generated feedback (binary correctness signals), we still tested the effect of feedback on the full LoCoMo dataset. Since LoCoMo requires static knowledge to answer questions, we only compared LLMsys equipped with memory

mechanisms. As shown in Table 17, most LLMsys variants demonstrate improved performance when feedback is available.

The comparison with and without feedback above and the off-policy experiments we report still differ in scope. Off-policy experiments additionally involve LLMsys storing and utilizing memory. To further investigate, we compare the performance of LLMsys on the training set under the settings of the off-policy experiments. Specifically, LLMsys is allowed to memorize the dialogues with feedback generated on the training set, but it is evaluated by answering questions from the same training set. At this time, the baseline Vanilla means the situation without feedback memory.

We report results for several LLMsys across three domains. In particular, since the Vanilla baseline cannot handle full static knowledge of LoCoMo and DialSim (in the Open domain), we exclude these two datasets when comparing the performance difference between LLMsys with and without feedback memory in the Open domain.

The results are shown in Table 18. Overall, LLMsys consistently outperforms Vanilla across all domain dataset categories, indicating that effective handling of procedural mem-

*Table 12.* Detailed per-domain evaluation results of LLMsys with Qwen3-8B backbone in the off-policy setting. For each dataset, we report raw performance scores using their dataset-specific evaluation metrics. Except for metrics marked with ↓, higher values indicate better performance.

| Domain | Dataset | Metric | Vanilla | BM25-M | BM25-S | Embed-M | Embed-S | A-Mem | Mem0 | MemoryOS |
|---|---|---|---|---|---|---|---|---|---|---|
| Open | LoCoMo | F1 | 0.3759 | 0.3795 | 0.4280 | 0.3056 | 0.5730 | 0.4148 | - | 0.4595 |
| | DialSim-friends | Accuracy | 0.1765 | 0.1765 | 0.4118 | 0.2941 | 0.1765 | 0.2941 | - | 0.0588 |
| | DialSim-bigbang | Accuracy | 0.3529 | 0.1176 | 0.2941 | 0.0588 | 0.3529 | 0.0588 | - | 0.0588 |
| | DialSim-theoffice | Accuracy | 0.3529 | 0.1176 | 0.3529 | 0.2353 | 0.4118 | 0.2353 | - | 0.0000 |
| | HelloBench-C.D. | Avg. Score | 0.8541 | 0.7220 | 0.8098 | 0.7892 | 0.8135 | 0.8223 | - | 0.7910 |
| | WritingPrompts | Meteor | 0.2337 | 0.2219 | 0.2356 | 0.2037 | 0.2284 | 0.2181 | - | 0.2270 |
| | WritingBench-C.D. | Score | 6.6860 | 6.4419 | 6.5233 | 6.4651 | 6.6279 | 6.6047 | - | 6.3372 |
| | NFCats | Score | 4.5800 | 3.9800 | 4.3600 | 4.1200 | 4.2400 | 4.5000 | - | 4.3200 |
| Academic | HelloBench-A.K.-QA | Avg. Score | 0.8364 | 0.8599 | 0.8423 | 0.8243 | 0.8724 | 0.8653 | 0.8451 | 0.8592 |
| | HelloBench-A.K.-Writing | Avg. Score | 0.8475 | 0.8353 | 0.8544 | 0.8544 | 0.8795 | 0.8446 | 0.8225 | 0.8392 |
| | IdeaBench | BERTScore | 0.5583 | 0.5590 | 0.5654 | 0.5598 | 0.5656 | 0.5568 | 0.5451 | 0.5585 |
| | | LLM Rating Score | 4.3000 | 4.3400 | 4.3000 | 4.2000 | 4.2200 | 3.8000 | 3.6600 | 4.4400 |
| | | LLM Novelty Ranking Score | 0.6600 | 0.5667 | 0.6400 | 0.5867 | 0.5667 | 0.5933 | 0.5333 | 0.4933 |
| | | LLM Feasibility Ranking Score | 0.1867 | 0.1200 | 0.2000 | 0.1600 | 0.1333 | 0.1467 | 0.0600 | 0.1200 |
| | SciTechNews | ROUGE-L | 0.1222 | 0.1210 | 0.1209 | 0.1236 | 0.1242 | 0.1149 | 0.1137 | 0.0900 |
| | | BERTScore-F1 | 0.8179 | 0.8185 | 0.8187 | 0.8181 | 0.8155 | 0.8173 | 0.8183 | 0.8110 |
| | | CLI↓ | 16.4420 | 16.5939 | 16.3955 | 16.6804 | 16.3926 | 16.5310 | 16.9725 | 15.3455 |
| | | FKGL↓ | 15.9355 | 16.2831 | 16.2497 | 16.5816 | 16.1097 | 16.2810 | 16.9436 | 14.9844 |
| | | DCRS↓ | 12.5779 | 12.6639 | 12.6515 | 12.7097 | 12.6230 | 12.6145 | 12.8170 | 12.3422 |
| | LimitGen-Syn | Accuracy | 0.4600 | 0.4800 | 0.4000 | 0.5200 | 0.4800 | 0.4600 | 0.5600 | 0.1200 |
| | | Rating | 1.3800 | 1.2400 | 1.3200 | 1.5400 | 1.5200 | 1.3800 | 1.5200 | 0.3200 |
| | WritingBench-A.E. | Score | 6.9118 | 6.5588 | 6.8529 | 6.5000 | 6.5000 | 6.8824 | 6.8235 | 4.9118 |
| Legal | JuDGE | Reasoning Meteor | 0.5138 | 0.4943 | 0.4996 | 0.4918 | 0.5025 | 0.4604 | 0.4823 | 0.4770 |
| | | Judge Meteor | 0.4077 | 0.4354 | 0.3425 | 0.4082 | 0.3868 | 0.4364 | 0.3559 | 0.3150 |
| | | Reasoning BERTScore | 0.8155 | 0.8134 | 0.8061 | 0.8015 | 0.8163 | 0.7601 | 0.8073 | 0.8092 |
| | | Judge BERTScore | 0.7897 | 0.7822 | 0.7410 | 0.7692 | 0.7710 | 0.7971 | 0.7693 | 0.7308 |
| | | Crime Recall | 0.9800 | 0.9600 | 0.9800 | 0.9800 | 0.9800 | 1.0000 | 0.9600 | 1.0000 |
| | | Crime Precision | 0.9600 | 0.9500 | 0.9600 | 0.9600 | 0.9600 | 0.9700 | 0.9500 | 0.9700 |
| | | Penalcode Index Recall | 0.7615 | 0.7504 | 0.7836 | 0.7402 | 0.7340 | 0.7352 | 0.7130 | 0.7195 |
| | | Penalcode Index Precision | 0.7293 | 0.7139 | 0.6964 | 0.7287 | 0.7213 | 0.7444 | 0.7471 | 0.7016 |
| | | Time Score | 0.7312 | 0.7009 | 0.6554 | 0.7005 | 0.6939 | 0.7060 | 0.6874 | 0.7026 |
| | | Amount Score | 0.4506 | 0.4224 | 0.4247 | 0.4437 | 0.4183 | 0.4892 | 0.4815 | 0.4796 |
| | LexEval-Summarization | ROUGE-L | 0.2206 | 0.2312 | 0.2259 | 0.2228 | 0.2271 | 0.2299 | 0.1961 | 0.2212 |
| | LexEval-Judge | ROUGE-L | 0.0644 | 0.1116 | 0.0864 | 0.0561 | 0.0863 | 0.0461 | 0.0461 | 0.0470 |
| | LexEval-QA | ROUGE-L | 0.0986 | 0.1359 | 0.1284 | 0.1240 | 0.1175 | 0.1335 | 0.1147 | 0.1009 |
| | WritingBench-P.L. | Score | 7.0732 | 6.8293 | 6.7805 | 6.7561 | 6.9268 | 6.7561 | 7.0000 | 5.9268 |

*Table 13.* Detailed evaluation results of LLMsys with Qwen3-8B backbone in the off-policy setting with both raw performance scores using their dataset-specific evaluation metrics where the value following the ± symbol represents the standard deviation across the test data. Except for metrics marked with ↓, higher values indicate better performance.

| Domain | Dataset | Metric | Vanilla | A-Mem | Mem0 | MemoryOS |
|---|---|---|---|---|---|---|
| **Open** | **LoCoMo** | **F1** | 0.3759±0.4232 | 0.4148±0.4709 | - | 0.4595±0.4846 |
| | **DialSim-friends** | **Accuracy** | 0.1765±0.3812 | 0.2941±0.4556 | - | 0.0588±0.2353 |
| | **DialSim-bigbang** | **Accuracy** | 0.3529±0.4779 | 0.0588±0.2353 | - | 0.0588±0.2353 |
| | **DialSim-theoffice** | **Accuracy** | 0.3529±0.4779 | 0.2353±0.4242 | - | 0.0000±0.0000 |
| | **HelloBench-C.D.** | **Avg. Score** | 0.8541±0.1301 | 0.8223±0.1490 | - | 0.7910±0.2006 |
| | **WritingPrompts** | **Meteor** | 0.2337±0.0434 | 0.2181±0.0588 | - | 0.2270±0.0503 |
| | **WritingBench-C.D.** | **Score** | 6.6860±1.2691 | 6.6047±1.4409 | - | 6.3372±1.5296 |
| | **NFCats** | **Score** | 4.5800±0.8964 | 4.5000±0.8307 | - | 4.3200±0.8352 |
| **Academic** | **HelloBench-A.K.-QA** | **Avg. Score** | 0.8364±0.1162 | 0.8653±0.0837 | 0.8451±0.1265 | 0.8592±0.1282 |
| | **HelloBench-A.K.-Writing** | **Avg. Score** | 0.8475±0.0654 | 0.8446±0.0738 | 0.8225±0.1062 | 0.8392±0.0775 |
| | **IdeaBench** | **BERTScore** | 0.5583±0.0336 | 0.5568±0.0338 | 0.5451±0.0330 | 0.5585±0.0340 |
| | | **LLM Rating Score** | 4.3000±2.1932 | 3.8000±2.1541 | 3.6600±1.9658 | 4.4400±2.1369 |
| | | **LLM Novelty Ranking Score** | 0.6600±0.4082 | 0.5933±0.4232 | 0.5333±0.4570 | 0.4933±0.4333 |
| | | **LLM Feasibility Ranking Score** | 0.1867±0.3138 | 0.1467±0.2918 | 0.0600±0.2180 | 0.1200±0.2473 |
| | **SciTechNews** | **ROUGE-L** | 0.1222±0.0213 | 0.1149±0.0278 | 0.1137±0.0194 | 0.0900±0.0273 |
| | | **BERTScore-F1** | 0.8179±0.0090 | 0.8173±0.0119 | 0.8183±0.0105 | 0.8110±0.0121 |
| | | **CLI↓** | 16.4420±1.5562 | 16.5310±1.5494 | 16.9725±1.5151 | 15.3455±2.1809 |
| | | **FKGL↓** | 15.9355±1.5586 | 16.2810±1.6296 | 16.9436±1.7064 | 14.9844±2.1855 |
| | | **DCRS↓** | 12.5779±0.5679 | 12.6145±0.6018 | 12.8170±0.6040 | 12.3422±1.2200 |
| | **LimitGen-Syn** | **Accuracy** | 0.4600±0.4984 | 0.4600±0.4984 | 0.5600±0.4964 | 0.1200±0.3250 |
| | | **Rating** | 1.3800±1.5085 | 1.3800±1.5861 | 1.5200±1.4730 | 0.3200±0.9042 |
| | **WritingBench-A.E.** | **Score** | 6.9118±1.8209 | 6.8824±1.6227 | 6.8235±1.6174 | 4.9118±2.8735 |
| **Legal** | **JuDGE** | **Reasoning Meteor** | 0.5138±0.1658 | 0.4604±0.1511 | 0.4823±0.1477 | 0.4770±0.1785 |
| | | **Judge Meteor** | 0.4077±0.2225 | 0.4364±0.2364 | 0.3559±0.1600 | 0.3150±0.1528 |
| | | **Reasoning BERTScore** | 0.8155±0.0529 | 0.7601±0.0876 | 0.8073±0.0452 | 0.8092±0.0639 |
| | | **Judge BERTScore** | 0.7897±0.0772 | 0.7971±0.0831 | 0.7693±0.0629 | 0.7308±0.1185 |
| | | **Crime Recall** | 0.9800±0.1400 | 1.0000±0.0000 | 0.9600±0.1960 | 1.0000±0.0000 |
| | | **Crime Precision** | 0.9600±0.1685 | 0.9700±0.1187 | 0.9500±0.2062 | 0.9700±0.1187 |
| | | **Penalcode Index Recall** | 0.7615±0.2329 | 0.7352±0.2522 | 0.7130±0.2633 | 0.7195±0.2741 |
| | | **Penalcode Index Precision** | 0.7293±0.2485 | 0.7444±0.2608 | 0.7471±0.2592 | 0.7016±0.2844 |
| | | **Time Score** | 0.7312±0.2297 | 0.7060±0.2681 | 0.6874±0.2444 | 0.7026±0.2446 |
| | | **Amount Score** | 0.4506±0.3583 | 0.4892±0.3642 | 0.4815±0.3583 | 0.4796±0.3783 |
| | **LexEval-Summarization** | **ROUGE-L** | 0.2206±0.0894 | 0.2299±0.0882 | 0.1961±0.0785 | 0.2212±0.0978 |
| | **LexEval-Judge** | **ROUGE-L** | 0.0644±0.1261 | 0.0461±0.1068 | 0.0461±0.0946 | 0.0470±0.1104 |
| | **LexEval-QA** | **ROUGE-L** | 0.0986±0.0710 | 0.1335±0.0614 | 0.1147±0.0625 | 0.1009±0.0691 |
| | **WritingBench-P.L.** | **Score** | 7.0732±1.3505 | 6.7561±1.3576 | 7.0000±1.4314 | 5.9268±2.7265 |

*Table 14.* The continued part of Table 13

| Domain | Dataset | Metric | BM25-M | BM25-S | Embed-M | Embed-S |
|---|---|---|---|---|---|---|
| Open | LoCoMo | F1 | 0.3795±0.4557 | 0.4280±0.4311 | 0.3056±0.3816 | 0.5730±0.4358 |
| | DialSim-friends | Accuracy | 0.1765±0.3812 | 0.4118±0.4922 | 0.2941±0.4556 | 0.1765±0.3812 |
| | DialSim-bigbang | Accuracy | 0.1176±0.3222 | 0.2941±0.4556 | 0.0588±0.2353 | 0.3529±0.4779 |
| | DialSim-theoffice | Accuracy | 0.1176±0.3222 | 0.3529±0.4779 | 0.2353±0.4242 | 0.4118±0.4922 |
| | HelloBench-C.D. | Avg. Score | 0.7220±0.2525 | 0.8098±0.1787 | 0.7892±0.2130 | 0.8135±0.1555 |
| | WritingPrompts | Meteor | 0.2219±0.0525 | 0.2356±0.0463 | 0.2037±0.0582 | 0.2284±0.0481 |
| | WritingBench-C.D. | Score | 6.4419±1.8018 | 6.5233±1.3786 | 6.4651±1.8783 | 6.6279±1.5704 |
| | NFCats | Score | 3.9800±1.2883 | 4.3600±1.0151 | 4.1200±1.0889 | 4.2400±1.0688 |
| Academic | HelloBench-A.K.-QA | Avg. Score | 0.8599±0.0870 | 0.8423±0.0935 | 0.8243±0.1835 | 0.8724±0.0781 |
| | HelloBench-A.K.-Writing | Avg. Score | 0.8353±0.0633 | 0.8544±0.0618 | 0.8544±0.0681 | 0.8795±0.0594 |
| | IdeaBench | BERTScore | 0.5590±0.0392 | 0.5654±0.0367 | 0.5598±0.0376 | 0.5656±0.0357 |
| | | LLM Rating Score | 4.3400±2.0553 | 4.3000±2.2023 | 4.2000±2.1633 | 4.3061±2.2150 |
| | | LLM Novelty Ranking Score | 0.5667±0.3844 | 0.6400±0.4155 | 0.5867±0.4246 | 0.5667±0.4435 |
| | | LLM Feasibility Ranking Score | 0.1200±0.2473 | 0.2000±0.3333 | 0.1600±0.2924 | 0.1333±0.3266 |
| | SciTechNews | ROUGE-L | 0.1210±0.0308 | 0.1209±0.0231 | 0.1236±0.0283 | 0.1242±0.0251 |
| | | BERTScore-F1 | 0.8185±0.0105 | 0.8187±0.0104 | 0.8181±0.0139 | 0.8155±0.0104 |
| | | CLI↓ | 16.5939±1.6383 | 16.3955±1.5075 | 16.6804±1.8093 | 16.3926±1.5199 |
| | | FKGL↓ | 16.2831±1.8239 | 16.2497±1.8087 | 16.5816±1.7954 | 16.1097±1.5616 |
| | | DCRS↓ | 12.6639±0.6876 | 12.6515±0.6562 | 12.7097±0.7112 | 12.6230±0.5916 |
| | LimitGen-Syn | Accuracy | 0.4800±0.4996 | 0.4000±0.4899 | 0.5200±0.4996 | 0.4800±0.4996 |
| | | Rating | 1.2400±1.4221 | 1.3200±1.7713 | 1.5400±1.5901 | 1.5200±1.6400 |
| | WritingBench-A.E. | Score | 6.5588±2.0174 | 6.8529±1.5743 | 6.5000±1.8511 | 6.5000±2.0037 |
| Legal | JuDGE | Reasoning Meteor | 0.4943±0.1585 | 0.4996±0.1657 | 0.4918±0.1688 | 0.5025±0.1539 |
| | | Judge Meteor | 0.4354±0.2211 | 0.3425±0.1785 | 0.4082±0.2172 | 0.3868±0.2012 |
| | | Reasoning BERTScore | 0.8134±0.0504 | 0.8061±0.0627 | 0.8015±0.0615 | 0.8163±0.0428 |
| | | Judge BERTScore | 0.7822±0.1361 | 0.7410±0.1264 | 0.7692±0.1359 | 0.7710±0.0701 |
| | | Crime Recall | 0.9600±0.1960 | 0.9800±0.1400 | 0.9800±0.1400 | 0.9800±0.1400 |
| | | Crime Precision | 0.9500±0.2062 | 0.9600±0.1685 | 0.9600±0.1685 | 0.9600±0.1685 |
| | | Penalcode Index Recall | 0.7504±0.2618 | 0.7836±0.2425 | 0.7402±0.2535 | 0.7340±0.2257 |
| | | Penalcode Index Precision | 0.7139±0.2466 | 0.6964±0.2466 | 0.7287±0.2324 | 0.7213±0.2522 |
| | | Time Score | 0.7009±0.2620 | 0.6554±0.2599 | 0.7005±0.2603 | 0.6939±0.2354 |
| | | Amount Score | 0.4224±0.3542 | 0.4247±0.3827 | 0.4437±0.3712 | 0.4183±0.3532 |
| | LexEval-Summarization | ROUGE-L | 0.2312±0.0944 | 0.2259±0.0941 | 0.2228±0.1000 | 0.2271±0.0830 |
| | LexEval-Judge | ROUGE-L | 0.1116±0.1535 | 0.0864±0.1488 | 0.0561±0.1158 | 0.0863±0.1473 |
| | LexEval-QA | ROUGE-L | 0.1359±0.0594 | 0.1284±0.0719 | 0.1240±0.0812 | 0.1175±0.0634 |
| | WritingBench-P.L. | Score | 6.8293±1.6064 | 6.7805±1.5852 | 6.7561±1.9099 | 6.9268±1.1974 |

*Table 15.* Off-policy experimental results with Qwen3-32B as the backbone model. The metric Average is the min-max normalization scores of each datasets and metric Z-Score is the average Z-Score.

| LLMsys | Open | | Academic | | Legal | |
|---|---|---|---|---|---|---|
| | **Average** | **Z-Score** | **Average** | **Z-Score** | **Average** | **Z-Score** |
| **Vanilla** | 0.6778 | 0.2917 | 0.7297 | 0.2205 | 0.4630 | -0.0061 |
| **BM25-M** | 0.6521 | 0.1743 | 0.7264 | 0.2352 | **0.4950** | **0.1504** |
| **BM25-S** | **0.7045** | **0.3289** | **0.7384** | 0.2593 | 0.4874 | 0.1250 |
| **Embed-M** | 0.6325 | 0.0809 | 0.7365 | 0.2366 | 0.4663 | 0.0342 |
| **Embed-S** | 0.6964 | 0.3149 | 0.7369 | **0.2736** | 0.4915 | 0.1265 |
| **A-Mem** | 0.6302 | 0.1041 | 0.7197 | 0.1557 | 0.4943 | 0.1497 |
| **Mem0** | - | - | 0.6896 | -0.0307 | 0.4739 | 0.0465 |
| **MemoryOS** | 0.6047 | -0.0293 | 0.6450 | -0.2241 | 0.4541 | -0.0247 |

ory and proper utilization of feedback can improve system performance. However, some LLMsys exhibit lower performance than Vanilla, suggesting that improperly leveraging memory, i.e., injecting irrelevant memory into the context, will degrade LLM generation quality.

### A.3.6. ON-POLICY EXPERIMENTS

We conducted on-policy experiments on each domains. For each dataset, we sampled instances and split them into training and test sets, which were then combined to form the training and test sets for each domain-level partition. The on-policy experimental procedure is as follows:

1. If a dataset contains static knowledge, we first load this knowledge into the memory of LLMsys.

2. At each on-policy step, we perform the following:

   (a) Randomly sample 100 training instances from the combined training set, and let LLMsys interact with the user feedback simulator to generate up to 3 rounds of dialogues.
   (b) Incorporate these 100 training dialogues into the memory of LLMsys, so that it accumulates more experience.
   (c) Evaluate the LLMsys with the updated memory on the entire test set and record its performance.

The generation parameters for the on-policy experiments are the same as those used in the off-policy experiments.

We report the performance of some LLMsys on domains Open, Academic, Legal, corresponding to Figs. 5, 6, 7, respectively.

### A.3.7. STEPWISE OFF-POLICY EXPERIMENTS

Following a similar setup to the on-policy experiments, we also conducted stepwise off-policy experiments. In this case,

we used the training dialogues generated previously instead of generating dialogues in real time. The stepwise off-policy procedure is as follows:

1. The training dialogues generated previously are grouped into batches of 100 instances.

2. At each step, we load one batch of dialogues into the memory of the LLMsys.

3. Evaluate the current LLMsys on the entire test set and record its performance.

The results of the off-policy experiments of some LLMsys on domains Open, Academic, Legal, corresponding to Figs. 8, 9 and 10, respectively.

### A.3.8. TIME CONSUMPTION ANALYSIS

To provide a more comprehensive view of the performance of LLMsys, we report the time consumption statistics in the task-partitioned off-policy experiments. Specifically, for each LLMsys, we measure the average time required to (i) memorize training dialogues and (ii) predict answers on the test set. All reported values are the average processing time per data point.

All experiments were conducted on an A800-80G GPU cluster. The backbone model Qwen3-8B was deployed via vLLM with 8-way GPU parallelism, where each GPU was utilized at approximately 80% capacity. In addition, we deployed Qwen3-Embedding-0.6B as the embedding model through vLLM on a single GPU (with 15% usage). For LLMsys that use Qwen3-Embedding-0.6B as the embedding model, embedding queries were routed through vLLM; in contrast, LLMsys that use all-MiniLM-L6-v2 ran the model directly on GPU without vLLM.

The detailed results are summarized in Table 19, which reports the average time per sample for both memory and

*Table 16.* Comparison of performance with and without feedback across different datasets. Results are reported with the dataset-specific evaluation metrics in the third column. With – Without denotes the performance gain achieved by incorporating feedback. Except for metrics marked with ↓, higher values indicate better performance.

| Dataset | #Sample/#Total | Metric | With Feedback | Without Feedback | With - Without |
|---|---|---|---|---|---|
| **JuDGE** | 2505 / 2505 | Reasoning Meteor | 0.5078 | 0.5080 | -0.0002 |
| | | Judge Meteor | 0.4510 | 0.4502 | **0.0008** |
| | | Reasoning BERTScore | 0.8106 | 0.8032 | **0.0074** |
| | | Judge BERTScore | 0.7918 | 0.7630 | **0.0288** |
| | | Crime Recall | 0.9557 | 0.9212 | **0.0345** |
| | | Crime Precision | 0.9552 | 0.9204 | **0.0348** |
| | | Crime F1 | 0.9554 | 0.9208 | **0.0346** |
| | | Penalcode Index Recall | 0.7518 | 0.7268 | **0.0250** |
| | | Penalcode Index Precision | 0.7166 | 0.6970 | **0.0196** |
| | | Penalcode Index F1 | 0.7338 | 0.7116 | **0.0222** |
| | | Time Score | 0.6968 | 0.6752 | **0.0216** |
| | | Amount Score | 0.5389 | 0.5206 | **0.0183** |
| **LexEval-Summarization** | 1000 / 1000 | ROUGE-L | 0.2650 | 0.2565 | **0.0085** |
| **LexEval-Judge** | 1000 / 1000 | ROUGE-L | 0.0441 | 0.0471 | -0.003 |
| **LexEval-QA** | 500 / 500 | ROUGE-L | 0.1008 | 0.0898 | **0.0110** |
| **WritingBench-A.E.** | 167 / 167 | Score | 6.7485 | 5.4072 | **1.3413** |
| **WritingBench-C.D.** | 422 / 422 | Score | 6.7393 | 5.346 | **1.3933** |
| **WritingBench-P.L.** | 201 / 201 | Score | 6.7811 | 5.9403 | **0.8408** |
| **WritingPrompts** | 2000 / 2000 | Meteor | 0.2358 | 0.2322 | **0.0036** |
| **SciTechNews** | 1000 / 1000 | ROUGE-L | 0.1195 | 0.1115 | **0.0080** |
| | | BERTScore-F1 | 0.8182 | 0.8141 | **0.0041** |
| | | CLI↓ | 15.3941 | 14.3085 | 1.0856 |
| | | FKGL↓ | 15.0427 | 14.2951 | 0.7476 |
| | | DCRS↓ | 12.0614 | 11.5219 | 0.5395 |
| **HelloBench-A.K.-QA** | 100 / 213 | Avg. Score | 0.8657 | 0.7425 | **0.1232** |
| **HelloBench-C.D.** | 100 / 228 | Avg. Score | 0.8737 | 0.6491 | **0.2246** |
| **HelloBench-A.K.-Writing** | 82 / 82 | Avg. Score | 0.8335 | 0.7787 | **0.0548** |
| **IdeaBench** | 100 / 2374 | BERTScore | 0.5574 | 0.5434 | **0.014** |
| | 100 / 2374 | LLM Rating Score | 4.7800 | 4.0900 | **0.6900** |
| | 100 / 2374 | LLM Novelty Ranking Score | 0.6033 | 0.2533 | **0.3500** |
| | 100 / 2374 | LLM Feasibility Ranking Score | 0.1967 | 0.1033 | **0.0934** |
| **LimitGen-Syn** | 100 / 1000 | Accuracy | 0.9500 | 0.9500 | **0.0000** |
| | | Rating | 2.6100 | 2.5900 | **0.0200** |
| **NFCats** | 100 / 2397 | Score | 4.4800 | 4.3700 | **0.1100** |

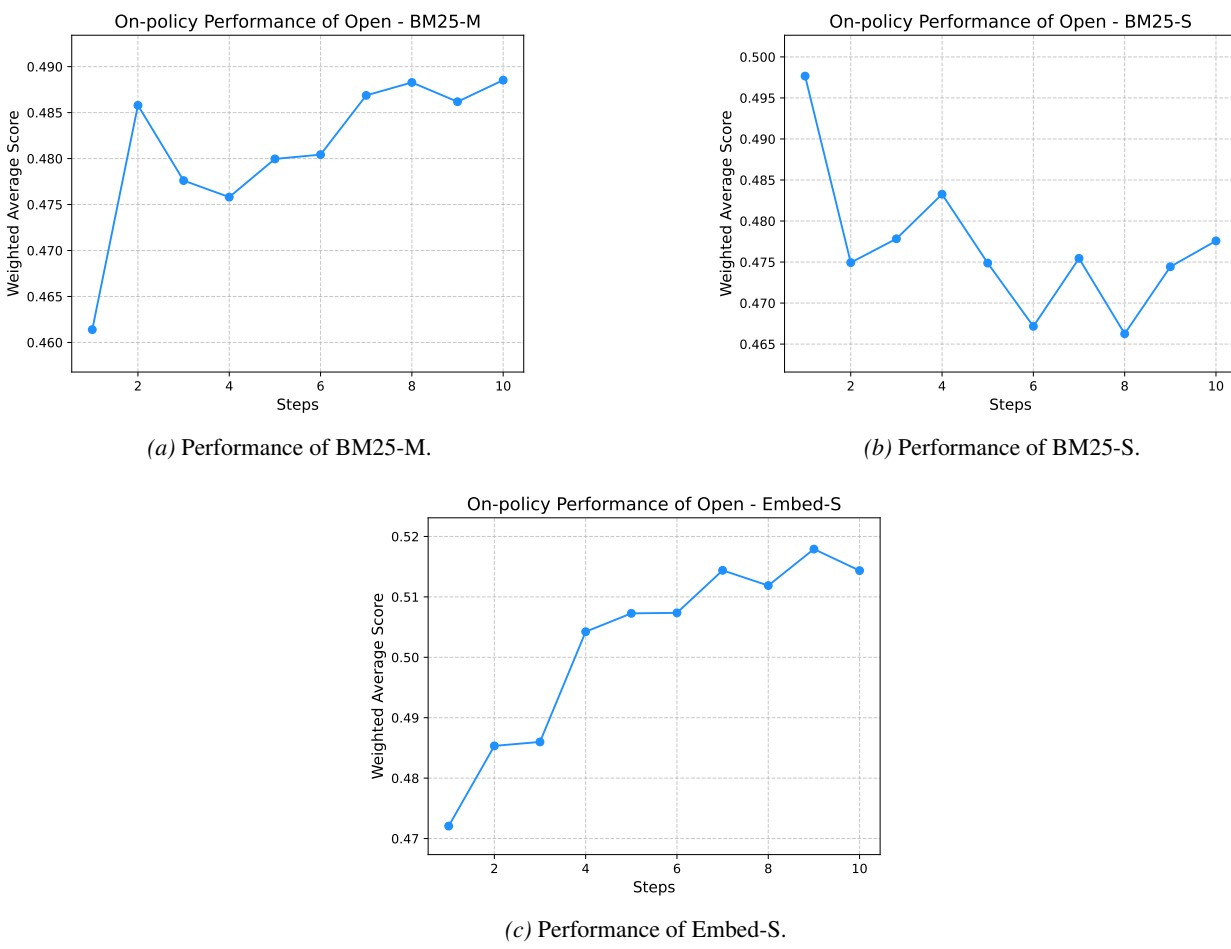

*(a)* Performance of BM25-M.         *(b)* Performance of BM25-S.

*(c)* Performance of Embed-S.

*Figure 5.* On-policy performance of some LLMsys on the Open domain. The x-axis denotes training steps, and the y-axis denotes the aggregated score.

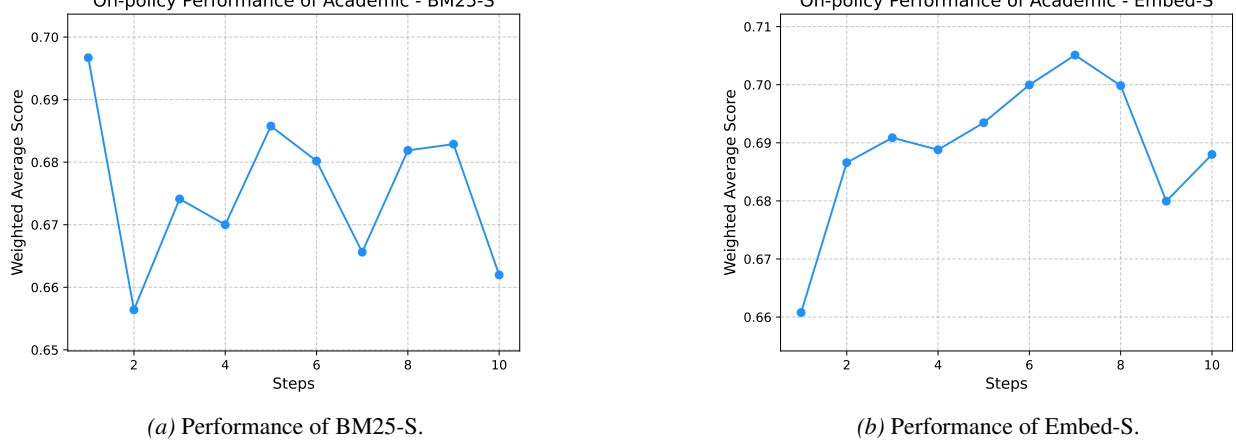

*(a)* Performance of BM25-S.         *(b)* Performance of Embed-S.

*Figure 6.* On-policy performance of some LLMsys on the Academic domain. The x-axis denotes training steps, and the y-axis denotes the aggregated score.

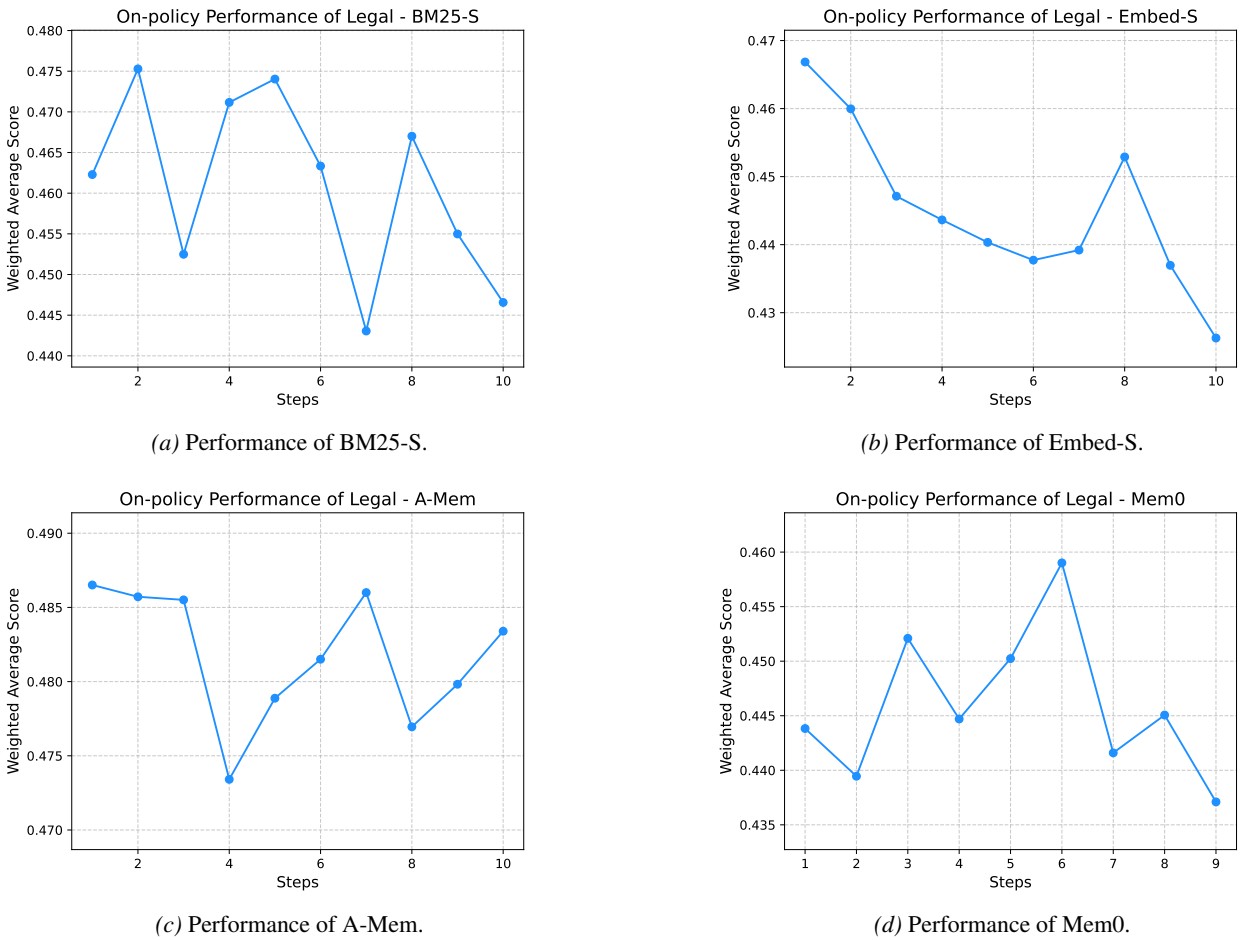

*(a)* Performance of BM25-S.

*(b)* Performance of Embed-S.

*(c)* Performance of A-Mem.

*(d)* Performance of Mem0.

*Figure 7.* On-policy performance of some LLMsys on the Legal domain. The x-axis denotes training steps, and the y-axis denotes the aggregated score.

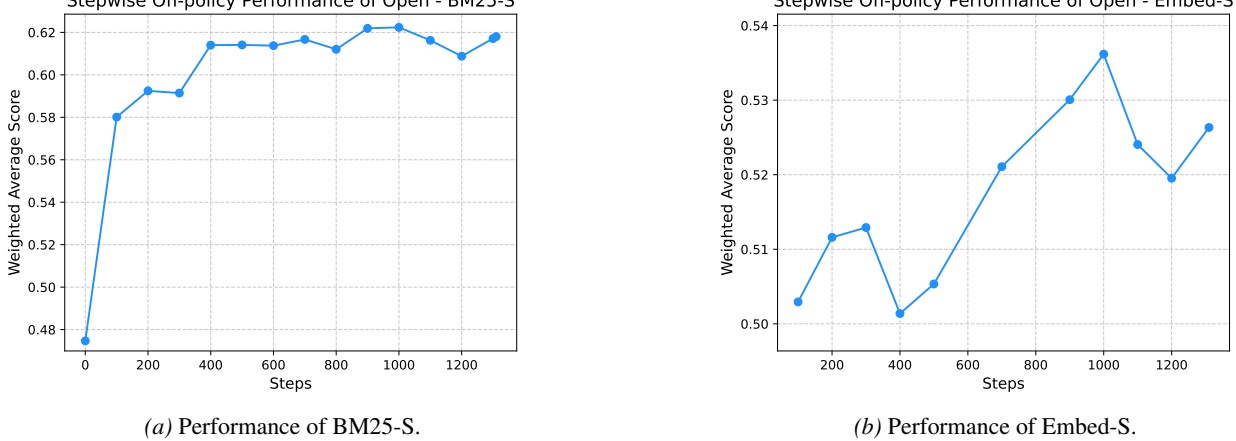

*(a)* Performance of BM25-S.

*(b)* Performance of Embed-S.

*Figure 8.* Stepwise off-policy performance of some LLMsys on the Open domain. The x-axis denotes training steps, and the y-axis denotes the aggregated score.

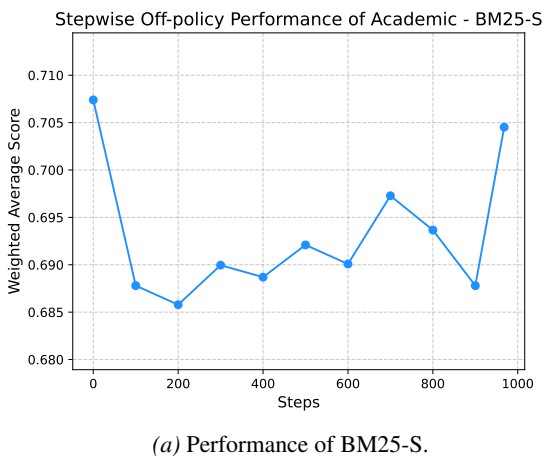

*(a)* Performance of BM25-S.

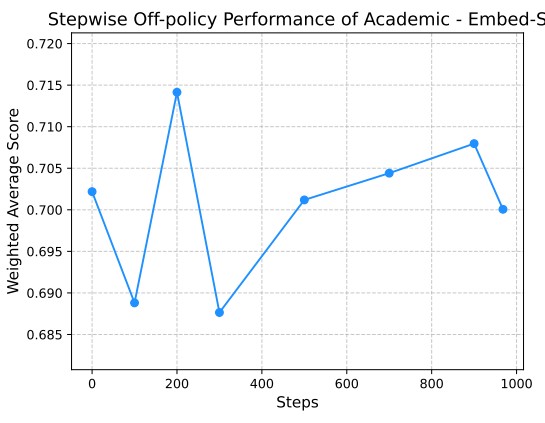

*(b)* Performance of Embed-S.

*Figure 9.* Stepwise off-policy performance of some LLMsys on the Academic domain. The x-axis denotes training steps, and the y-axis denotes the aggregated score.

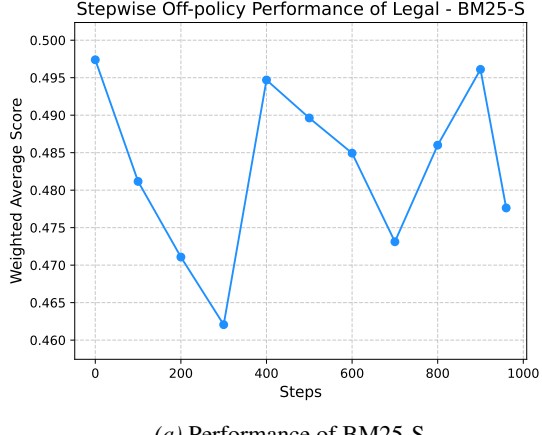

*(a)* Performance of BM25-S.

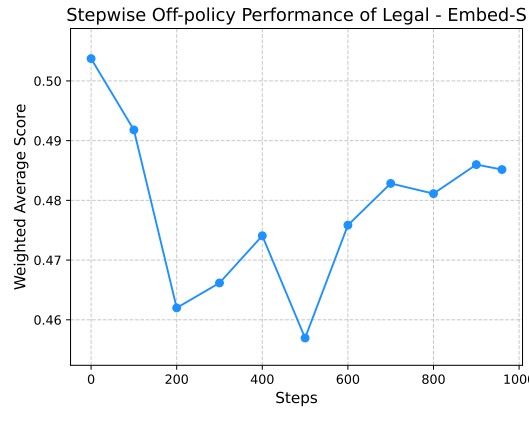

*(b)* Performance of Embed-S.

*Figure 10.* Stepwise off-policy performance of some LLMsys on the Legal domain. The x-axis denotes training steps, and the y-axis denotes the aggregated score.

*Table 17.* Performance of LLMsys on the full Locomo dataset with and without feedback. Results are reported using the F1 score as the evaluation metric. With – Without denotes the performance gain achieved by incorporating feedback.

| LLMsys | With Feedback | Without Feedback | With - Without |
|--------|---------------|------------------|----------------|
| **BM25-M** | 0.3249 | 0.3256 | -0.0007 |
| **BM25-S** | 0.4358 | 0.4273 | **0.0085** |
| **Embed-M** | 0.4310 | 0.4272 | **0.0038** |
| **Embed-S** | 0.4679 | 0.4650 | **0.0029** |
| **A-Mem** | 0.3954 | 0.3674 | **0.0280** |
| **MemoryOS** | 0.2710 | 0.2724 | -0.0014 |

*Table 18.* Off-policy performance of several LLMsys on training sets across three major dataset domains. Open is denoted as Open* because Locomo and DialSim are excluded. Each entry reports two numbers in the form a / b, where a is the average min-max normalization scores and b is the z-score.

| LLMsys | Open* | Academic | Legal |
|--------|-------|----------|-------|
| **Vanilla** | 0.8003 / 0.2481 | 0.7215 / 0.1923 | 0.4671 / 0.0025 |
| **BM25-M** | 0.7559 / 0.0247 | 0.6862 / 0.0048 | **0.4775 / 0.0523** |
| **BM25-S** | 0.7972 / 0.2335 | **0.7269 / 0.2135** | 0.4719 / 0.0374 |
| **Embed-M** | 0.7478 /-0.0188 | 0.6916 /-0.0034 | 0.4507 /-0.0730 |
| **Embed-S** | **0.8045 / 0.269** | 0.7124 / 0.1583 | 0.4680 / 0.0228 |
| **A-Mem** | 0.7813 / 0.1493 | 0.7199 / 0.1820 | 0.4694 / 0.0438 |

prediction operations. We also provide visualizations in Figure 4 for clearer comparisons.

Overall, we observe that MemoryOS exhibits a significantly higher memory cost than all other LLMsys, with its average memorization time exceeding all baselines by a large margin. Mem0 ranks second in terms of memory latency, though it is still substantially slower than the other five memory-based systems. The memory efficiency of A-Mem is close to that of the RAG systems while maintaining comparable predicting efficiency. These findings highlight a key trade-off in memory systems: while more complex memory mechanisms can enhance capability, they often come at the cost of substantially increased memory operation latency.

A closer comparison between Mem0 and MemoryOS reveals an interesting discrepancy. While the results in Table 19 show that MemoryOS generally incurs much higher memory latency than Mem0, their behaviors diverge when dealing with static knowledge from DialSim and LoCoMo. The knowledge is pre-collected conversations. Following the official LoCoMo implementation, each message within a dialogue is stored as an independent memory entry for both Mem0 and MemoryOS.

In this setting, MemoryOS demonstrated extremely fast performance. For example, memorizing all 2,347 sessions in DialSim-theoffice required only 0.4937 seconds, and memorizing the first conversation in LoCoMo (19 sessions in total) took merely 0.0125 seconds. In contrast, Mem0 was con-

siderably slower: storing the first conversation in LoCoMo took more than 6 minutes. Moreover, Mem0 completely failed to finish memorizing DialSim-theoffice. After running for six hours, we observed a dramatic slowdown. As shown in Figure 11(a), the x-axis represents the cumulative number of entries memorized, and the y-axis indicates the elapsed memorization time, with one point plotted for every 20 entries on average. The visualization clearly shows that while Mem0 initially maintains a stable memorization speed, the time per entry increases drastically in the later stages, eventually exceeding 20 minutes per entry. This pathological slowdown persisted even after 12 hours of execution, at which point we had to terminate the process manually. Similarly, we observed the same phenomenon when using Mem0 to memorize dialogues in the Legal domain that included copy feedback. After a certain period, the memorization time of Mem0 increased sharply. Even when memorization succeeded, the reasoning speed remained extremely slow, preventing normal completion of inference. Figure 4(c) also illustrates Mem0's unusually slow inference speed compared to it on other tasks.

An ideal memory system should be able to adapt its storage strategy to the structure of the incoming data. In our current implementations, memory modules are restricted to storing either plain text inputs or dialogues in a relatively fixed format. However, real-world usage scenarios often involve more complex structures. For example, a dialogue can naturally be decomposed into multiple entries, which would

better support fine-grained retrieval; at the same time, the entire dialogue should also be preserved as a single entry to ensure contextual integrity. Designing memory systems that can flexibly process and store diverse input modalities or hierarchical data remains an important direction for future research.

### A.3.9. ACTION FEEDBACK

In addition to text feedback, we also explored other types of feedback, such as like and copy feedback. Since the number of examples with likes or copy actions is relatively small, for each dataset, we did not use the same training set defined in the off-policy experimental setting. Instead, with the same test set in the off-policy experiments, we considered all remaining data points and selected dialogues containing like or copy feedback as the training dialogues. We conducted experiments in both the Academic and Legal domains. In the Academic domain, there are 517 dialogues with like feedback and 137 dialogues with copy feedback. In the Legal domain, there are 394 dialogues with like feedback and 1,200 dialogues with copy feedback. Apart from using a different set of training dialogues for memorization of LLMsys, all other settings are identical to those in the off-policy experiments.

Specifically, we also explored using SFT as a baselinefee. Since SFT does not handle text feedback well, we actually filtered for dialogues in which the user feedback simulator was satisfied with (like), or wanted to copy, the LLM system's response in the first turn. Note that these dialogues are different from the training dialogues used above. For training, we used the first turn of each dialogue (i.e., the user and assistant's initial exchange) as input, with the assistant's response serving as the training target. In this filtering setting, the Academic domain contains 257 first-turn dialogues with like feedback and 84 with copy feedback, while the Legal domain contains 221 with like feedback and 737 with copy feedback. We trained LoRA adapters of Qwen3-8B on these data separately, with the following configuration: learning rate of 1e-4, 5 epochs, batch size of 4, gradient accumulation steps of 1, and input truncation length of 8192. The LoRA parameters were set as $r = 8, \alpha = 32$ and dropout=0.05.

The final experimental results are shown in Table 20. For the setting that uses training dialogues with copy feedback in the Legal domain, although the Mem0 can successfully memorize these dialogues after a very long time (over 10 hours, as shown in Figure 11(b)), the inference time becomes strangely slow (about 22min/case), so we did not complete this experiment.

**Noisy Action Feedback** In real-world scenarios, user feedback from human often contains noise, such as user

preferences that may not align with each situation. To investigate the robustness of LLMsys under varying levels of noise, we introduce noise into user action feedback using two different methods.

The first method preserves the original user satisfaction scores while introducing noise by adjusting the probability of giving like feedback at different score levels. This is controlled by tuning the hyperparameter $S_{0L}$ in the formula 1. Specifically, when $S_{0L}$ is high, the user simulator is more selective; the probability of giving a "like" is highly concentrated on high scores (e.g., scores 9-10), and low scores are unlikely to giving a "like". When $S_{0L}$ is low, the user is more lenient; the "like" probability distribution is more uniform, and medium-to-low scores also have a reasonable probability of giving a "like". As shown in Figure 12, the probability curves differ significantly when $S_{0L}$ is set to 3 versus 9.

We evaluate the performance of several LLMsys on the Legal domain under different values of $S_{0L}$, and the results are shown in Figure . In this type of noisy action feedback, the two RAG systems exhibit relatively small performance fluctuations and perform better when $S_{0L}$ is high, suggesting that RAG systems prefer higher-quality data to enhance generation. In contrast, A-Mem is more sensitive to noise: its performance peaks at $S_{0L} = 6.5$, decreases when $S_{0L}$ is too low, and drops sharply when $S_{0L}$ is too high. This This indicates that A-Mem may benefit more from broader data coverage rather than strictly optimal data. Mem0 does not demonstrate a clear trend.

The second method maintains the original score–like probability mapping while introducing Gaussian perturbations to the user satisfaction scores. The like feedback is then reassessed based on the perturbed scores. Here, $\sigma$ denotes the standard deviation of the Gaussian perturbation: larger $\sigma$ values result in greater deviations in satisfaction and, consequently, higher levels of noise. To illustrate the effect of $\sigma$ more intuitively, using a fixed random seed of 42, we generated 100000 satisfaction scores from 1 to 10 and applied Gaussian perturbations with different $\sigma$ values. Table 21 shows how many scores changed and the total amount of score variation caused by these perturbations.

We evaluate the performance of several LLMsys on the Legal domain under different perturbation levels $\sigma$, with results shown in Figure 14. Under Gaussian-noise conditions, Embed-S and A-Mem are strongly affected: their performance decreases significantly as $\sigma$ increases. In contrast, BM25-S and Mem0 are less sensitive to noise. BM25-S exhibits a slight performance drop with increasing $\sigma$, while Mem0 shows a performance decline at $\sigma = 0.8$ and $\sigma = 1.0$, followed by a modest recovery when $\sigma = 1.2$.

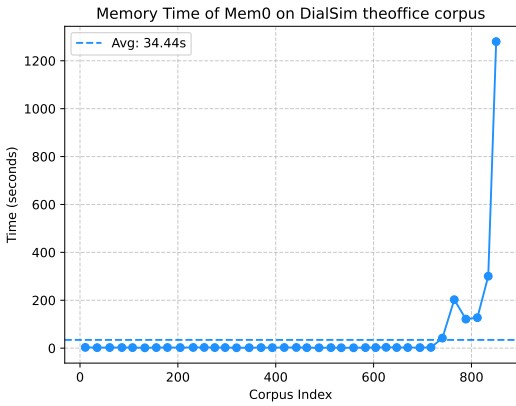
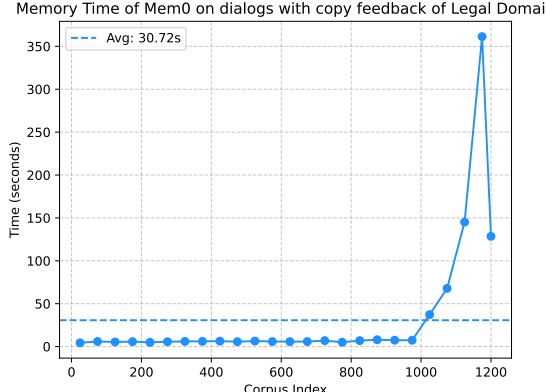

*(a)* Memorizing the first 854 sessions of DialSim-theoffice corpus. One point is plotted for every 20 entries on average.

*(b)* Memorizing all dialogues with `copy` feedback of Legal domain. One point is plotted for every 50 entries on average.

*Figure 11.* Mem0 exhibits severe slowdown during memorization across different tasks. (a) For memorizing the corpus of DialSim-theoffice, the time per entry grows drastically after an initially stable phase, eventually exceeding 20 minutes per entry. (b) A similar pathological slowdown is observed when memorizing dialogues with `copy` feedback in Legal domain.

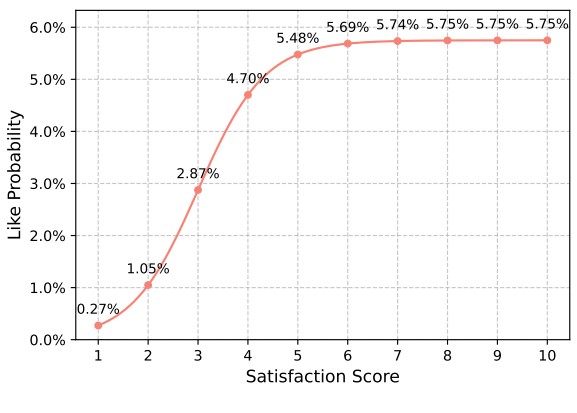
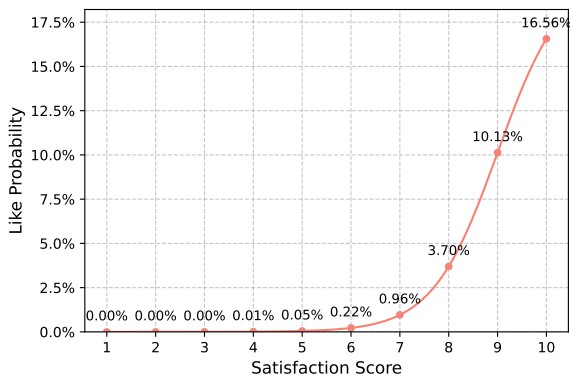

*(a)* Like probability on $S_{0L} = 3$.

*(b)* Like probability on $S_{0L} = 9$.

*Figure 12.* Probability of giving like feedback at different satisfaction scores under two noise levels. Figure (a) shows the like–score mapping when $S_{0L}$ is low, resulting in a more uniform probability across scores. Figure (b) shows the mapping when $S_{0L}$ is high, where the probability mass becomes highly concentrated on high scores.

### A.3.10. EFFECT OF BACKBONES FOR LLMSYS AND USER SIMLUATION

To further analyze whether the choice of LLMsys and user simluation backbones would affect the relative performance of different baselines, we conduct experiments on Legal domain with Mistral3.2-24B[7] in the off-policy settings. Results are reported in Table 22. As shown in the table, Mistral3.2-24B outperformed Qwen3-8B when used as LLMsys backbones, which is not surprising considering that their model sizes. Yet, the relative performance orders of different LLMsys are mostly the same. BM25-M is still the best method, and almost all RAG baselines outperform

the SOTA memory-based methods. When comparing the results using Qwen3-32B vs. Mistral3.2-24B as the user simulator, we observe the same trend on LLMsys (all using Qwen3-8B as the backbone). This indicates that the performance differences between different LLMsys are caused by their design, not their backbones or the backbones of the user simulators.

### A.3.11. CASE STUDY

To further illustrate why memory-based baselines performed differently with RAG methods, we conducted a case study on Legal domain and provide an example with the task question, ground truth, metric performance, as well as the memory entries retrieved by the LLMsys and their final responses in Table 23. Overall, none of the methods managed

---

[7]https://huggingface.co/mistralai/Mistral-Small-3.2-24B-Instruct-2506

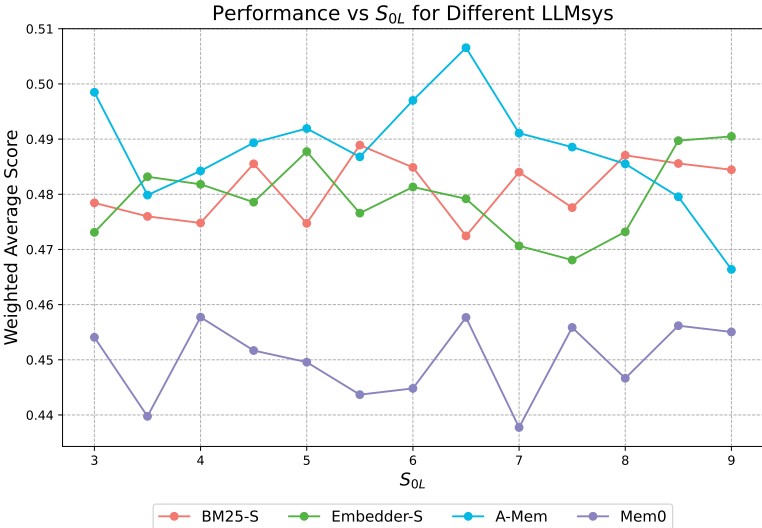

*Figure 13.* Performance of different LLMsys under noisy like feedback with varying $S_{0L}$ on the Legal domain.

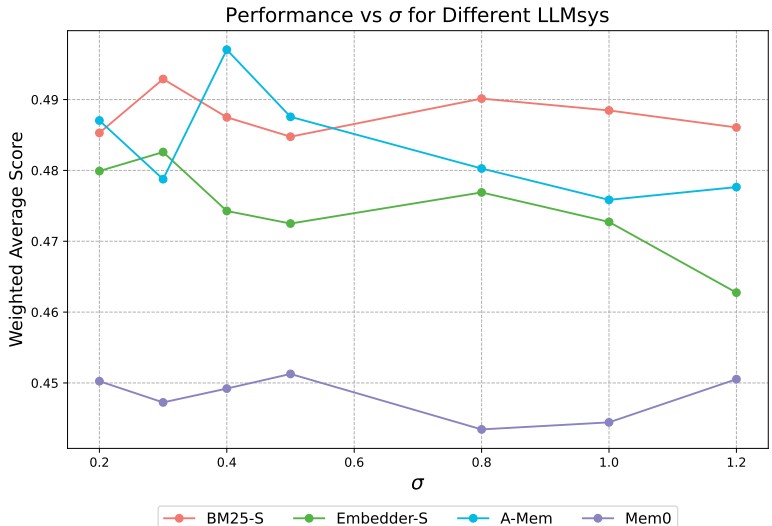

*Figure 14.* Performance of different LLMsys under noisy like feedback with varying $S_{0L}$ on the Legal domain.

to retrieve memory entries that directly address the question and thus their Rouge-L scores are not high. However, the entries retrieved by BM25-S are more relevant to legal topics and involve less disrupting tokens such as "Speaker", "Assistant", "User asked about", etc. This might be the reason why it produces better response with higher Rouge-L. From this case, we can see that the performance of existing memory-based LLMsys are far from perfect, which means that more efforts on developing better architectures and algorithms to utilize different types of context and memory information are needed.

*Table 19.* Time consumption of different LLMsys in the off-policy experiments. The reported values are the average time per sample for memory operations (Memory Time) and prediction on the test set (Predict Time), measured in iterations per second.

| Task | LLMsys | Memory Time (it/s) | Predict Time (it/s) |
|------|--------|--------------------|---------------------|
| LiSo | Vanilla | - | 0.4677 |
|      | BM25-M | 0.0246 | 7.6561 |
|      | BM25-S | 0.0362 | 5.7793 |
|      | Embed-M | 0.2196 | 0.6319 |
|      | Embed-S | 0.2076 | 0.8796 |
|      | A-Mem | 0.2368 | 0.7820 |
|      | Mem0 | - | - |
|      | MemoryOS | 17.7130 | 0.7336 |
| SiLo | Vanilla | - | 1.7714 |
|      | BM25-M | 0.3380 | 1.8759 |
|      | BM25-S | 0.1406 | 2.6643 |
|      | Embed-M | 0.2043 | 1.9481 |
|      | Embed-S | 0.1119 | 2.6131 |
|      | A-Mem | 0.1677 | 2.1971 |
|      | Mem0 | 3.9274 | 1.9500 |
|      | MemoryOS | 15.2478 | 3.2378 |
| SiSo | Vanilla | - | 1.1567 |
|      | BM25-M | 0.3193 | 1.1119 |
|      | BM25-S | 0.1562 | 1.5833 |
|      | Embed-M | 0.2000 | 1.2121 |
|      | Embed-S | 0.1096 | 1.5435 |
|      | A-Mem | 0.1775 | 1.3525 |
|      | Mem0 | 2.4689 | 1.1061 |
|      | MemoryOS | 18.2458 | 2.0068 |
| LiLo | Vanilla | - | 2.2757 |
|      | BM25-M | 0.4331 | 8.9927 |
|      | BM25-S | 0.5467 | 5.5455 |
|      | Embed-M | 0.2315 | 2.7985 |
|      | Embed-S | 0.1691 | 3.5934 |
|      | A-Mem | 0.2319 | 3.4343 |
|      | Mem0 | 4.6919 | 27.3889 |
|      | MemoryOS | 21.3488 | 3.0327 |

*Table 20.* We report results on the Academic and Legal domains, using training dialogues with like or copy feedback. For each system, both the min-max normalization average score and its corresponding Z-score are shown. The training dialogues are selected based on user actions (like or copy) rather than the off-policy training set, while the test set remains the same as in the off-policy experiments.

| LLMsys | Academic | | | | Legal | | | |
|---|---|---|---|---|---|---|---|---|
| | like | | copy | | like | | copy | |
| | Average | Z-Score | Average | Z-Score | Average | Z-Score | Average | Z-Score |
| **Vanilla** | 0.7154 | 0.1450 | **0.7154** | **0.1450** | 0.4637 | -0.0019 | 0.4637 | -0.0019 |
| **SFT** | **0.7193** | **0.1719** | 0.7139 | 0.1331 | 0.4832 | 0.0929 | 0.4817 | 0.0702 |
| **BM25-M** | 0.6882 | 0.0126 | 0.6844 | 0.0058 | **0.5024** | **0.1694** | 0.4782 | 0.0447 |
| **BM25-S** | 0.6910 | 0.0476 | 0.6548 | -0.0751 | 0.4799 | 0.0633 | **0.4853** | **0.0907** |
| **Embed-M** | 0.6880 | -0.0126 | 0.6942 | 0.0052 | 0.4692 | 0.0114 | 0.4639 | -0.0074 |
| **Embed-S** | 0.6950 | 0.0780 | 0.6588 | -0.0680 | 0.4873 | 0.1114 | 0.4762 | 0.0555 |
| **A-Mem** | 0.7112 | 0.1327 | 0.6860 | -0.0249 | 0.4892 | 0.1343 | 0.4623 | 0.0243 |
| **Mem0** | 0.6733 | -0.1042 | 0.6780 | -0.0855 | 0.4362 | -0.1324 | - | - |
| **MemoryOS** | 0.7065 | 0.1134 | 0.6600 | -0.1183 | 0.4455 | -0.0826 | 0.4226 | -0.1753 |

| $\sigma$ | Change Rate | Total Score Variation |
|---|---|---|
| **0.2** | 1.20% | 1197 |
| **0.3** | 8.69% | 8691 |
| **0.4** | 18.84% | 18853 |
| **0.5** | 28.59% | 28809 |
| **0.8** | 47.86% | 52752 |
| **1** | 55.56% | 67169 |
| **1.2** | 60.76% | 80503 |

*Table 21.* The impact of Gaussian perturbations on user satisfaction scores. Larger standard deviations ($\sigma$) introduce more noise.

*Table 22.* Off-policy experiments on Legal domain (explicit verbal feedback only, min-max normalization) with different backbones for LLMsys and user simulator.

| User Simulator | **Qwen3-32B** | | **Mistral3.2-24B** |
|---|---|---|---|
| LLMsys Backbone | **Qwen3-8B** | **Mistral3.2-24B** | **Qwen3-8B** |
| Vanilla | 0.4637 | 0.4743 | 0.4637 |
| BM25-M | **0.5011** | **0.5215** | **0.4943** |
| BM25-S | 0.4776 | 0.5137 | 0.4815 |
| Embed-M | 0.4635 | 0.5059 | 0.4546 |
| Embed-S | 0.4806 | 0.5207 | 0.4814 |
| A-Mem | 0.4684 | 0.4738 | 0.4655 |
| Mem0 | 0.4451 | 0.4734 | 0.4381 |
| MemoryOS | 0.4290 | 0.4696 | 0.4410 |

*Table 23.* An example test case sampled from the Legal domain. The official metric used by this test case is Rouge-L. The Retrieved Memory is the memory entry/document retrieved by the LLMsys based on the user question, and the Response is the final ouput generated by the LLMsys.

| User Question |
|---|
| Wang (a judge at City A Court) was investigated by the Supervisory Committee for bribery and malfeasance. The prosecution filed a public suit against Wang at City A Court... Question: Are there any problems regarding jurisdiction in this case? Please explain. |

| Ground Truth |
|---|
| The trial jurisdiction exercised by the People's Court of City A contravenes the provisions of the law. According to Article 16 of the Interpretation of the Criminal Procedure Law: "Where a competent People's Court, due to circumstances such as the necessity for ... |

| Method | Rouge-L | Retrieved Memory | Response |
|---|---|---|---|
| Mem0 | 0.1111 | 1. Sun knowingly accepted criminal proceeds (concealing crime)... 2. User asked about Chen's meritorious service (ruled out due to official duties)... | Wang's case being prosecuted at City A Court is legal and compliant... Moreover, the behavior of Sun appropriating 170,000 yuan from the company, as mentioned, should be evaluated based on both the source of the funds and his subjective intent... |
| BM25-S | 0.1749 | 1. Analyzed the criminal liability of an accomplice (Liu) in a blackmail case, discussing "control theory" vs. "loss of control theory" regarding the point of completion and the concept of "inheriting accomplice."... 2. Discussed the procedure when a new crime (intentional injury) is discovered after a death penalty order is signed but before execution, requiring suspension and Supreme Court review... | Analyzed a case involving a judge (Wang) accused of bribery and malfeasance to determine if there were jurisdictional errors. The response concluded that there were no jurisdictional issues, citing the Criminal Procedure Law and Supervision Law. It validated the Supervisory Commission's authority to investigate duty crimes, the prosecution's indictment process, the court's jurisdiction to try the case, and the legality of the separate prosecution of the nephew for concealing criminal proceeds... |
| A-Mem | 0.1630 | 1. Analyzed the invalidity of an arbitration clause in an administrative asset acquisition agreement... 2. Discussed whether facts confirmed in an arbitration award exempt a party from the burden of proof in subsequent civil litigation... | Analyzed the case of Judge Wang (accused of bribery and malfeasance) to determine if there were jurisdictional errors. The response concluded that there were no jurisdictional issues, validating (1) The Supervisory Commission's authority to investigate duty crimes... |
| MemoryOS | 0.1333 | 1. Assistant provided legal analysis on joint marital debt at 2023-10-05 ... Assistant explained legal procedures for death penalty execution ... 2. Assistant provided legal analysis on administrative litigation scope ... | ... Yes, the prosecutor can directly prosecute Wang Xiaoliu... The prosecutor generally can directly file a public prosecution without prior filing for investigation or detection... based on Article 175... provided the facts are clear and evidence is sufficient ... Therefore, in Wang's case, the prosecutor can directly prosecute Wang Xiaoliu without separate investigation... |

