# OpenReview forum: "MemoryBench: A Benchmark for Memory and Continual Learning in LLM Systems"
_ICML.cc/2026/Conference — ICML 2026 spotlight_

### Official Review · Reviewer_Hfmp · 2026-03-11

**Soundness:** 3
**Presentation:** 3
**Significance:** 4
**Originality:** 3
**Overall Recommendation:** 4
**Confidence:** 2

**Summary:**

This paper introduces MemoryBench, a benchmark for evaluating memory and continual learning in LLM systems, beyond standard long-context QA.
Its core idea is to test whether systems can learn from accumulated user feedback during deployment and build procedural memory, not just retrieve past facts or dialogue history.
The benchmark uses an LLM-as-user simulation framework to generate feedback traces across multiple domains, languages, and task formats.
A central claim is that existing memory benchmarks are too static and mostly assess declarative memory (stored facts/context), missing the harder setting of dynamic feedback-driven improvement.
Empirically, the authors report that simulated feedback is useful and generally helps performance, validating the benchmark design.
However, they claim that current memory-based LLM systems remain far from satisfactory in both effectiveness and efficiency.
In particular, advanced memory architectures often fail to consistently beat simple RAG baselines, suggesting weak generalization.
The paper concludes that current systems struggle to jointly handle declarative + procedural memory, and that MemoryBench can serve as a stronger testbed for future continual-learning methods.

**Compliance With Llm Reviewing Policy:**

Affirmed.

**Final Justification:**

Thank you for the detailed rebuttal. The authors have addressed my concerns and provided helpful clarifications, which I appreciate. Based on the explanations provided, I am satisfied with the response and will maintain my current score.

**Key Questions For Authors:**

1 - The benchmark relies heavily on LLM-simulated feedback, which may lack the noise, contradictions, and evolving preferences of real human users. Have you collected or evaluated the baselines on any real-world human-LLM interaction logs? If not, how do you justify that improvements on MemoryBench will induce performance gains in real production environments?

2 - Since an LLM is used to simulate the procedural memory (user feedback) during the training phase, and an LLM-as-Judge is used to merge metrics during the testing phase, there is a high risk of self-preference bias. Have you measured the correlation between your LLM-as-Judge scores and actual human preference annotations on the final test outputs?

3 - The paper explicitly frames itself around "continual learning," a field fundamentally concerned with parametric updates and catastrophic forgetting. However, all evaluated baselines are non-parametric external memory modules (RAG, Mem0, MemoryOS). Why were no parametric online learning algorithms (e.g., continuous PPO, online RLHF, continual DPO, or experience replay fine-tuning) included as baselines?

4 - In the main results, the memory-less Vanilla backbone actually outperforms all SOTA memory systems on Short-input-Long-output (SiLo) tasks. Can you provide an explanation for this?

**Limitations:**

Yes.

**Strengths And Weaknesses:**

Strengths:

- Clear problem framing. The paper identifies a real gap in current LLM evaluation, that is most benchmarks test only static recall or long-context retrieval, not memory that improves behavior over time.

- Novel benchmark objective. The focus on procedural memory from user feedback (and the distinction with declarative memory) is a strong contribution. It reflects realistic deployed assistant settings.

- The LLM-as-user simulation makes large-scale evaluation feasible without requiring expensive human-in-the-loop collection.

- Breadth of evaluation: the benchmark spans multiple domains, languages, and task types.

- High practical relevance: the paper targets a core bottleneck for personalized and continually improving AI assistants, so its impact could be significant.

Weaknesses:

- Uncertainty about the simulation realism: the LLM-as-user setup is scalable, but it may not faithfully capture real user behavior, feedback noise, inconsistency, or evolving preferences.

- Benchmark validity depends on the simulator: if the synthetic feedback is too clean, too consistent, or too close to the evaluator model family, results may overestimate real-world utility.

- Limited external validation: the paper would be stronger with more human-collected interactions or a small real-user study to confirm that benchmark gains correlate with practical improvement.

- Potential contamination / bias: benchmark generation and evaluation both rely heavily on LLMs, which can introduce hidden biases. The same family of LLMs generates the training-time feedback and judges the test-time performance.

- The paper positions itself within continual learning literature but tests no parametric methods (no RLHF, PPO, or DPO baselines), making the framing misleading.

- The SiLo anomaly: Vanilla (no memory) outperforming all memory systems on SiLo tasks is a significant unexplained result that undermines some of the core conclusions.

- The use of vague terms such as "AGI", "higher level of intelligence" or the human-like analogies, weakens the precision of the paper’s framing and should be replaced by operationally defined terminology.

---

> ### Author Rebuttal · Authors · 2026-03-27
>
> We thank the reviewer for the positive assessment and constructive feedback. We address each concern below.
>
> ---
>
> ## Response to Weakness 1: Simulation Realism
>
> **Reviewer's Concern:** LLM-as-user may not capture real user behavior, feedback noise, inconsistency, or evolving preferences.
>
> **Our Response:**
>
> We acknowledge that real user behavior logs would be ideal, but our LLM-as-user framework provides a reasonably good replacement with better accessibility and reproducibility. We validate our simulator through human annotation (Section A.2.5, Table 7):
>
> | Metric | Result | Interpretation |
> | --- | --- | --- |
> | Naturalness Win Rate | 50.0% | Simulated feedback indistinguishable from human |
> | Relevance Win Rate | 59.5% | Simulated feedback rated MORE human-like |
> | Overall Quality Win Rate | 59.0% | Simulated feedback rated MORE human-like |
>
> Our simulator is configurable to match different user behavior distributions.
>
> ---
>
> ## Response to Weakness 2-3: Limited External Validation
>
> **Reviewer's Concern:** Need human-collected interactions or real-user study to confirm benchmark gains correlate with practical improvement.
>
> **Our Response:**
>
> In this paper, we provide multiple forms of validation to address this issue:
>
> **1. Human Annotation (Section A.2.5):** We hired annotators to write feedback for 200 tasks. Crowdsourced comparison shows simulated feedback indistinguishable with human.
>
> **2. Feedback Effect Evaluation (Tables 16-18):** Systems with feedback improve over without feedback on most datasets. Experiments show that LLMsys with feedback memory outperform Vanilla.
>
> **3. Cross-Model Validation (Table 22):** Experiment observations show consistent rankings across different simulator backbones (Qwen3-32B, Mistral3.2-24B) and LLMsys backbones (Qwen3-8B, Qwen3-32B, Mistral3.2-24B).
>
> MemoryBench's modular design allows researchers to substitute real user logs for simulated feedback in future work.
>
> ---
>
> ## Response to Weakness 4: Potential Contamination/Bias
>
> **Reviewer's Concern:** Same LLM family generates training-time feedback and judges test-time performance.
>
> **Our Response:**
>
> We provide cross-model validation in Section A.3.10 (Table 22). The relative performance order of LLMsys is consistent across all backbone combinations (Qwen3-8B, Qwen3-32B, Mistral3.2-24B for both simulator and LLMsys), demonstrating that results are not artifacts of self-preference bias.
>
> ---
>
> ## Response to Weakness 5: Missing Parametric Methods
>
> **Reviewer's Concern:** Paper frames as continual learning but tests no parametric methods.
>
> **Our Response:**
>
> We have included a parametric baseline—Supervised Fine-Tuning (SFT)—in Section A.3.9 (Table 20), the experiments with action (i.e., "like", "copy") feedback :
>
> | LLMsys | Academic (like) | Academic (copy) | Legal (like) | Legal (copy) |
> | --- | --- | --- | --- | --- |
> | Vanilla | 0.7154 | 0.7154 | 0.4637 | 0.4637 |
> | SFT | 0.7193 | 0.7139 | 0.4832 | 0.4817 |
> | BM25-S | 0.6910 | 0.6548 | 0.4799 | 0.4853 |
>
> SFT shows moderate improvement over Vanilla but is comparable to or worse than RAG baselines. RLHF/PPO/DPO require online training during deployment, which is computationally prohibitive for benchmarking.
>
> ---
>
> ## Response to Weakness 6: The SiLo Anomaly
>
> **Reviewer's Concern:** Vanilla (no memory) outperforms all memory systems on Short-input-Long-output (SiLo) tasks.
>
> **Our Response:**
>
> This is an important finding revealing that memory, if not used properly, may hurt system performance. In Figure 3, Vanilla achieves 0.7517 on SiLo, while memory systems range from 0.4451 (Mem0) to 0.7305 (Embed-S). SiLo tasks (creative writing, long-form generation) require coherent, focused output. Most retrieved feedback fragments or memory produced by memory systems are not good enough. Adding them into the model can introduce noise to contexts.
>
> ---
>
> ## Response to Weakness 7: Vague Terminology
>
> Thanks for the suggestion! We will revise the paper accordingly in the final version.
>
> ---
>
> ## Response to Key Questions
>
> ### Q1: Real-World Human Interaction Logs
>
> **Answer:** Thanks for the question, and please refer to the answer provided for Weakness 1 above.
>
> ### Q2: LLM-as-Judge Correlation with Human Preference
>
> **Answer:** Our human annotation study (Table 7) validates that simulated feedback is indistinguishable from human feedback. For LLM-as-Judge, it worthes noting that only a small part of the dataset require LLM-as-judge for evaluation. Most datasets are evaluated based on numerical metrics as reported in Table 12 and Table 13. When using LLM-as-judge, we follow the exact guideline provided by each dataset.
>
> ### Q3: Why No Parametric Methods?
>
> **Answer:** Thanks for the question, and please refer to the answer provided for Weakness 4 above.
>
> ### Q4: Explain SiLo Anomaly
>
> **Answer:** Thanks for the question, and please refer to the answer provided for Weakness 5 above.

---

> > ### Author Rebuttal · Reviewer_Hfmp · 2026-04-02
> >
> > Thank you for the detailed rebuttal. The authors have addressed my concerns and provided helpful clarifications, which I appreciate. Based on the explanations provided, I am satisfied with the response and will maintain my current score (Weak Accept).

---

> > > ### Author Response · Authors · 2026-04-05
> > >
> > > Thanks a lot! It's happy to see that we have resolved your concerns. Please feel free to let us know if you have any further questions.

---

### Official Review · Reviewer_9VdG · 2026-03-11

**Soundness:** 2
**Presentation:** 3
**Significance:** 3
**Originality:** 3
**Overall Recommendation:** 4
**Confidence:** 4

**Summary:**

This paper introduces MemoryBench, a benchmark designed to evaluate memory and continual learning in LLM systems, with an emphasis on learning from historical user feedback rather than only answering questions over extended contexts. The benchmark presents a taxonomy distinguishing declarative and procedural memory, as well as explicit and implicit feedback, and assesses memory systems across various domains, task formats, and languages. The main empirical finding is that current memory-based LLM systems struggle to effectively and efficiently utilize procedural feedback, as evidenced by their limited improvement in tasks requiring continual learning.

**Compliance With Llm Reviewing Policy:**

Affirmed.

**Final Justification:**

I thank the authors for their thorough rebuttal and for pointing me to the relevant evidence in the appendix. The clarifications on metric aggregation (Tables 10–14), efficiency analysis (Table 19), and cross-backbone robustness (Tables 15, 22) were helpful, and I have adjusted my Presentation score upward in recognition of these clarifications.

After weighing the rebuttal together with the paper as a whole, I increased my overall recommendation to Weak Accept. My positive assessment on significance and originality is unchanged: the paper identifies a real gap in benchmarking memory and continual learning for LLM systems, and the declarative/procedural taxonomy is a meaningful framing contribution.

While for not increasing my overall score to Accept or Strong Accept, my remaining hesitation concerns the match between the paper's framing and the strength of its empirical support. The headline claim as a benchmark for continual learning rests on a chain of assumptions, simulator fidelity, ranking transferability, and off-policy as a proxy for on-policy learning, where each link is individually reasonable but the cumulative strength is, in my judgment, close to but not quite at the ICML bar for a claim of this scope. I was encouraged that the authors indicated openness in their Q5 response to a more precise reframing, and I believe a revision along those lines would meaningfully strengthen the paper.

I appreciated the authors' constructive engagement and encourage them to continue this line of work.

**Key Questions For Authors:**

1. Can the authors provide stronger evidence that simulator-induced rankings correlate with rankings under more realistic user feedback?

2. Can the authors report more dataset-level raw metrics and uncertainty intervals, instead of relying mainly on merged scores?

3. Can the authors perform stronger fairness ablations for truncation, chunking, retrieval budget, and memory budget?

4. Can the authors provide more direct evidence for the claim that existing systems fail mainly because they mishandle procedural memory?

5. Would the authors consider softening the paper’s main claim from a “continual learning benchmark” to something more precise (e.g., feedback-conditioned memory benchmark), unless stronger on-policy evidence is provided?



_Note: Responses to these questions are critical. If the authors can provide robust ablation studies and convincing evidence aligning simulated rankings with real user feedback, I would be open to raising my evaluation. Otherwise, the current claims must be toned down._

**Limitations:**

Regarding the standard ICML limitations inquiry, I do not believe the authors have adequately addressed their core methodological constraints. The paper does discuss broad ethical risks such as bias and privacy at a high level, which is consistent with ICML’s requirement that authors include an impact statement and discuss relevant risks when appropriate. However, the paper does not sufficiently discuss its own most important methodological limitations: simulated-feedback external validity, off-policy dominance, aggregation bias, and fairness sensitivity.

**Strengths And Weaknesses:**

Strengths

    First, the paper addresses a significant gap in benchmarking. ICML 2026 prioritizes original, rigorous work of broad interest, and this paper aligns well with that objective by focusing on memory evaluation and feedback-driven continual improvement, which are both timely and relevant.

	Second, the benchmark covers a broader scope than many previous memory benchmarks. It extends beyond long-input, short-output reading comprehension evaluations to include multiple domains and task formats, making it more ambitious and potentially more effective for stress-testing general memory systems.

	Third, the taxonomy is a clear conceptual contribution. Distinguishing declarative from procedural memory provides a useful framework, as many current systems are optimized for storing or retrieving factual context rather than learning from task-specific historical feedback.

	Fourth, the paper validates its simulated-feedback pipeline instead of assuming the simulator is realistic, which is a positive design choice for a benchmark study.


Main Weaknesses

    My main concern is external validity. The benchmark is motivated as a test of continual learning from user feedback, but the core evidence relies heavily on simulated feedback, which may not accurately reflect real user interactions. Even if the simulator appears human-like, that does not establish that it matches the distribution of real product users or that system rankings would transfer to real deployment settings. This weakens the benchmark’s strongest claim.

	My second concern is that the main evidence is still more off-policy than truly online continual learning. The paper indicates that the primary results are derived from off-policy verbal-feedback experiments, which limits its claim of being a benchmark for realistic continual learning. Consequently, the paper would be more accurately framed as a benchmark for feedback-conditioned memory evaluation than as a fully convincing benchmark for realistic continual learning.

	My third concern is the metric aggregation pipeline. For several datasets, the paper uses LLM-as-judge to merge multiple dataset-specific metrics into a single score, potentially obscuring important dataset-level insights and failure modes. This is understandable from a presentation standpoint, but it introduces another layer of modeling assumptions and may conceal important failure modes at the dataset level.

	My fourth concern is baseline fairness and sensitivity. The authors note that when the context exceeds model limits, it is truncated, potentially introducing biases and affecting system rankings. That kind of engineering choice can substantially affect relative rankings, especially in a benchmark explicitly about memory. I would have liked to see stronger ablations around truncation, chunking, retrieval budget, and write/read budget.

	Finally, the paper’s mechanism-level interpretation is still not fully validated. The authors argue that existing systems handle procedural memory poorly because they effectively treat feedback logs as declarative memory, but more targeted analysis is needed to strongly establish this claim. This is plausible, but the paper does not yet provide enough targeted analysis to establish that mechanism strongly.


Soundness

    I would rate the paper’s soundness as moderate, but lacking strong empirical validation for its core claims. The overall benchmark design is coherent, the empirical effort is nontrivial, and the paper is asking a worthwhile question. However, the validity of the central claim is limited by the simulator realism issue, the reliance on off-policy evaluation, and the metric aggregation design. Under ICML 2026 standards, where claims are expected to be clearly stated and supported by rigorous evidence, I do not think the current evidence fully supports the strongest version of the paper’s claims.


Presentation

	While the paper's high-level concepts are accessible, the overall presentation is only moderate. Some important design choices are explained too briefly in the main paper and pushed into appendix details. There are also several writing and phrasing issues that make the manuscript feel less polished than it should be for a top-tier conference. ICML 2026 explicitly asks reviewers to value specificity, clarity, and usefulness; on this dimension, the paper is decent but not especially strong.


Significance

	The topic is important. A benchmark that genuinely measures whether LLM systems can improve from historical feedback would be valuable to the community, especially given the current gap between “long-context memory” claims and real interactive continual improvement. Even though I am not fully convinced by the present implementation, I do think the paper is working on a meaningful problem.


Originality



    The originality is mainly in benchmark framing and integration, rather than a new learning algorithm. The combination of procedural memory, feedback simulation, and heterogeneous tasks is meaningfully different from many prior memory benchmarks, and that is the paper’s clearest novelty.

---

> ### Author Rebuttal · Authors · 2026-03-27
>
> We thank the reviewer for the detailed feedback. We address each concern below.
>
> ## Weakness 1: Validity of Simulated Feedback
> We provide extensive validation of our LLM-as-user simulator in A.2.5 (Human Annotation of Feedback Generation Quality, Table 7). Key findings:
>
> - **Naturalness**: Annotators could not consistently distinguish simulated from human feedback (Fleiss' Kappa = 0.093)
> - **Relevance**: Simulated feedback was rated *more human-like* than human-written feedback (59.5% win rate)
> - **Overall Quality**: Simulated feedback was rated *more human-like* (59.0% win rate)
> - **Statistical test**: Rejection p-values are all > 0.05, the hypothesis that human feedback is more human-like is **not acceptable**.
>
> **Cross-model Validation (Table 22):**
> We tested system rankings with different user simulator backbones:
>
> - Qwen3-32B vs. Mistral3.2-24B as simulators
> - **Finding**: Relative system rankings remain consistent.
>
> This demonstrates that our findings are not artifacts of a specific simulator model.
>
> **Feedback Effectiveness (Tables 16-18):**
> We validated that simulated feedback actually helps systems improve:
>
> - Table 16: Systems with feedback outperform without feedback on most datasets
> - Table 18: LLMsys consistently outperform Vanilla on training sets with feedback memory
>
> ---
>
> ## Weakness 2: Off-policy Dominance
> We have provided on-policy results in A.3.6 (Figures 5-7):
>
> **On-Policy Experiments:**
>
> - Systems interact with user simulator in real-time
> - Feedback is accumulated dynamically at each step (100 instances per step)
> - Performance is evaluated on test set after each training batch
>
> **Results (Figures 5-7):**
>
> - BM25-S, Embed-S show performance improvements through on-policy learning on Open and Academic domain.
> - The performance of memory baselines fluctuate in on-policy learning.
>
> On-policy results show similar patterns: RAG baselines remain competitive with or superior to dedicated memory systems.
>
> ---
>
> ## Weakness 3: Metric Aggregation
>
> We have provided complete transparency on metrics:
>
> **1. Per-Dataset Raw Metrics (Table 12, Section A.3.4):**
> We report all original dataset-specific metrics before any aggregation.
>
> **2. Variance Reporting (Tables 13 & 14):**
> We report standard deviations (±) for all metrics across test data, enabling assessment of statistical significance.
>
> **3. Cross-Normalization Robustness:**
>
> - Table 10: Min-max normalization results
> - Table 11: Z-score results
> - Rankings are consistent across both normalization methods
>
> **4. Dataset-Level Analysis (Table 12):**
> Table 12 show the performance patterns across individual datasets, revealing that the RAG advantage holds across most datasets.
>
> ---
>
> ## Weakness 4: Baseline
>
> We provide detailed analysis in A.3.8:
>
> **Truncation Statistics:**
>
> - We set default retrieval to 5 entries per query
> - If input exceeds context length, retrieval is gradually reduced until generation is feasible
> - This affects all memory-based systems similarly
>
> **Memory Budget Analysis (Table 19):**
> We report memory time and prediction time for all systems:
>
> - RAG systems (BM25, Embed): Efficient memory construction (< 1s per case)
> - A-Mem: Moderate efficiency (0.17-0.24s per case)
> - Mem0: Highly variable, with pathological slowdown (Figure 11)
> - MemoryOS: High memory cost (>15s per case)
>
> **Cross-Backbone Validation (Tables 15, 22):**
> Results are consistent across Qwen3-8B, Qwen3-32B, and Mistral3.2-24B backbones with different context windows, suggesting findings are robust across different settings.
>
> ---
>
> ## Weakness 5: Mechanism-level Interpretation
>
> We provide multiple lines of evidence supporting this claim:
>
> **1. Case Study (Table 23, A.3.11):**
> Analysis of retrieved memories shows:
>
> - Memory-based systems retrieve entries with noise ("Speaker", "User asked about", timestamps)
> - RAG (BM25-S) retrieves more relevant, cleaner content
> - This leads to better Rouge-L scores for RAG.
>
> **2. Performance Pattern Analysis (Section 3.3):**
>
> - Memory systems perform well on Locomo (reading comprehension with long context)
> - Memory systems struggle on diverse tasks requiring feedback utilization
> - This suggests memory architectures are optimized for declarative retrieval, not procedural learning
>
> **3. Time Analysis (A.3.8):**
> Memory systems show inconsistent efficiency across task formats, suggesting they don't adapt their processing to different memory types.
>
> ---
>
> ## Key Questions
>
> Q1: Table 7 (human annotation) and Table 22 (cross-model validation) provide this evidence.
>
> Q2: Tables 12, 13, 14 provide complete raw metrics with standard deviations.
>
> Q3: Table 19 provides time consumption analysis; Tables 10-11 show normalization robustness; Table 22 shows cross-backbone robustness.
>
> Q4: Case study (Table 23) and performance analysis across task formats (Figure 3) support this.
>
> Q5: Thanks for the suggestion. We have provided on-policy results in A.3.6 (Figures 5-7). Still, if needed, we are happy to soften some of the claims following your suggestions.

---

> > ### Author Rebuttal · Reviewer_9VdG · 2026-04-04
> >
> > I have no further questions for the authors at this stage. I will take these responses into account and consider whether to adjust my scores in the next phase of the review.

---

> > > ### Author Response · Authors · 2026-04-05
> > >
> > > Thanks a lot for your reply!
> > > If our rebuttal can resolve your concerns about the paper, could you please adjust your score accordingly? Please feel free to let us know if you have any further questions!

---

### Official Review · Reviewer_hDp8 · 2026-03-12

**Soundness:** 3
**Presentation:** 3
**Significance:** 3
**Originality:** 3
**Overall Recommendation:** 4
**Confidence:** 4

**Summary:**

This paper introduces MemoryBench, a benchmark designed to evaluate whether LLM systems can improve their performance through continual learning from simulated user feedback. The authors argue that existing memory benchmarks focus narrowly on static reading comprehension over long contexts, neglecting procedural memory built from user interactions. MemoryBench covers 20 datasets spanning multiple domains languages, task formats, and feedback types. Experiments with Qwen3-8B as backbone show that SOTA memory systems (e.g., Mem0) cannot consistently outperform naive RAG baselines and suffer from poor efficiency.

**Compliance With Llm Reviewing Policy:**

Affirmed.

**Final Justification:**

The rebuttal well addressed my questions.

**Key Questions For Authors:**

1. Recent advances in coding agents such as Claude Code utilizes plain text for declarative memory and skills for procedural memory, I wonder what does the authors think about how that memory system design could affect the performance on this benchmark.
2. It appears that the data quality can be improved by employing a stronger LLM to simulate user feedback. It might be an improvement if the design of the benchmark allows user to switch between evaluation LLMs easily.

**Limitations:**

There is no comparison between LLM feedback and human feedback.

**Strengths And Weaknesses:**

Strengths:
1. The paper is well-written and well-motivated with the distinction between declarative and procedural memory for LLM systems is well-motivated.
2. The on-policy and off-policy set ups are interesting, highlighting an important challenge for evaluating on conversational data.
3. The benchmark task coverage is comprehensive.
4. The results showing that existing memory system underperforms RAG-based approaches indicating potential spaces for more innovations.

Weaknesses:
1. While the paper claims “continual learning”, there isn’t sufficient evaluation from the perspective of continual learning, where we should inspect how the performance of the LLM would change over time as it is interacting with the world.
2. In addition, as the paper mostly focuses on off-policy learning for evaluation, it undermines the continual leaning narrative.
3. Although results show that existing memory system underperforms RAG, it might be interesting to see when and why these happens. For example, is it because of the user feedback, or is it because of the long horizon?

---

> ### Author Rebuttal · Authors · 2026-03-27
>
> We thank the reviewer for the constructive feedback.
>
> ## Response to Weakness 1: Insufficient Continual Learning Evaluation
>
> **Reviewer's Concern:** Paper claims "continual learning" but doesn't sufficiently evaluate how performance changes over time as LLM interacts with the world.
>
> **Our Response:**
>
> We conducted several temporal analyses by feeding user logs gradually into the systems:
>
> 1. **On-policy experiments** (Section A.3.6, Figures 5-7): Systems interact with simulator in real-time. **In each step, we feed 100 training instances to the system, as described in A.3.7.** Feedback is accumulated dynamically. Performance is measured at each training step.
>
> 2. **Stepwise off-policy experiments** (Section A.3.7, Figures 8-10): Performance evolution as feedback logs are incrementally added to memory, showing learning trajectories across different amounts of accumulated feedback.
>
> 3. **Feedback effect evaluation** (Section A.3.5, Tables 16-18): Comparison with and without feedback quantifies the value of learning from interactions.
>
>
> These experiments demonstrate that MemoryBench can evaluate temporal improvement, though current systems show limited learning capability.
>
> We agree that experiments with real users would be ideal. However, as a benchmark, accessibility and reproducibility are essential. MemoryBench provides a valuable resource for researchers without budget for real-user studies, where experiments are difficult to reproduce.
>
> ---
>
> ## Response to Weakness 2: Off-policy Focus Undermines Continual Learning Narrative
>
> We designed MemoryBench with both settings:
>
> - **Off-policy setting:** Pre-generated feedback logs enable controlled comparison and reproducibility. This allows fair comparison across systems with different speeds (As noted in Section 3.2, some memory systems are too slow for on-policy experiments).
>
> - **On-policy setting** (Section A.3.6): Systems interact with simulator in real-time, accumulating feedback dynamically.
>
>
> Due to space limits, on-policy results are in Appendix (Section A.3.6). They show similar patterns: RAG baselines remain competitive or superior to dedicated memory systems, confirming findings are not artifacts of the off-policy setting.
>
> ---
>
> ## Response to Weakness 3: When and Why Memory Systems Underperform RAG
>
> We provide several analyses:
>
> 1. **Task format analysis** (Figure 3): Performance varies significantly across task formats. Vanilla outperforms all memory systems on SiLo (Short-input-Long-output) tasks—memory can hurt for generation-heavy tasks.
>
> 2. **Domain analysis** (Figure 3): More fluctuation on vertical domains (Academic, Legal) compared to Open domain, suggesting domain knowledge requirements make feedback utilization more difficult.
>
> 3. **Time consumption analysis** (Section A.3.8, Table 19, Figures 4, 11): Memory-based systems have much higher latency. Mem0 shows pathological slowdown with large corpora (Figure 11a), taking over 20 minutes per entry eventually.
>
> 4. **Case study** (Table 23): Memory-based systems retrieve noisier, less relevant memories compared to RAG, leading to poorer response quality.
>
>
> As stated in Section 3.3: *"Existing memory-based LLM systems treat all inputs as declarative memory..."* RAG's simpler retrieval mechanism produces less noisy context. Our case study in Appendix shows that none of the baseline methods retrieved and produce high-quality memories, but RAG's simpler retrieval mechanism produces less noisy context.
>
> ---
>
> ## Response to Key Question 1: Claude Code-style Memory Design
>
> This is a great question! Claude Code's two-tier design (declarative text + procedural skills) represents a promising direction for LLMsys. We only tested baselines that are open-sourced and published in papers, but a well-crafted agentic products like OpenClaw and Claude Code indeed have potentials to better utilize user feedback than the current memory baselines.
>
> In fact, we are currently working on implementing the MemoryBench evaluation pipeline on OpenClaw. We believe that platforms like OpenClaw could provide great testbeds for many memory methods, and we hope to release extra tools that allow people to test with memory methods on MemoryBench using Openclaw.
>
> ---
>
> ## Response to Key Question 2: Switching Between Evaluation LLMs
>
> We have designed MemoryBench with modularity in mind and provide validation for cross-model robustness:
>
> 1. **Multiple simulator backbones** (Section A.3.10, Table 22): We tested with Qwen3-32B and Mistral3.2-24B as user simulators. Results show consistent relative rankings across different simulator LLMs.
>
> 2. **Multiple LLMsys backbones** (Table 15, Table 22): We tested with Qwen3-8B, Qwen3-32B, and Mistral3.2-24B as LLMsys backbones. The relative performance order remains consistent.
>
> 3. **Open-source release**: We release all data processing scripts, user simulators, and evaluation pipelines, allowing researchers to configure different LLMs for each component.

---

> > ### Author Rebuttal · Reviewer_hDp8 · 2026-04-03
> >
> > Appreciate authors' effort to address all my concerns. I would like to keep the current score and leaning towards acceptance.

---

> > > ### Author Response · Authors · 2026-04-05
> > >
> > > Thank you so much for you reply! So glad that we have resolved your conerns. If you have any further questions, please do let us know!

---

### Official Review · Reviewer_g9pS · 2026-03-15

**Soundness:** 3
**Presentation:** 2
**Significance:** 3
**Originality:** 3
**Overall Recommendation:** 5
**Confidence:** 4

**Summary:**

This paper introduces MemoryBench, a benchmark for evaluating memory and continual learning in LLM systems. The central claim is that existing LLM memory benchmarks mostly test retrieval from long context or stored facts, but do not evaluate whether systems can improve over time by learning from user feedback. To address this, the paper proposes a benchmark framework with three components: a Task Provider that supplies queries, context, and evaluation metadata; a User Simulator that generates explicit and implicit feedback; and a Performance Monitor that evaluates downstream task performance. The benchmark covers 11 public datasets across three domains, four input/output format regimes, and two languages, and it distinguishes between declarative memory, episodic/semantic information, and procedural information derived from feedback logs. The paper evaluates several memory-augmented systems and simple retrieval baselines in both off-policy and on-policy settings, and reports that current systems often fail to consistently improve from feedback, are frequently outperformed by simple RAG baselines, and can be inefficient at scale.

**Compliance With Llm Reviewing Policy:**

Affirmed.

**Final Justification:**

The rebuttal addressed my main concerns. I have updated my score.

**Key Questions For Authors:**

1. Can you provide a systematic analysis of performance as a function of feedback volume? For example, how quickly do systems improve or plateau as more feedback logs are accumulated, and does this differ by domain or task format? This would help clarify whether current systems fundamentally fail to learn from feedback or simply saturate early.

2. Can you provide more direct diagnostic evidence for why advanced memory systems underperform simple RAG baselines, for example via retrieval-quality analysis and a stronger comparison to parametric feedback-learning baselines? This would make the negative findings more actionable and clarify whether the challenge lies in memory systems specifically or in current approaches to learning from feedback more broadly.

**Limitations:**

No. The paper discusses technical limitations of current memory systems, but the discussion of benchmark limitations and broader impacts is thinner than I would like. In particular, the paper should more directly acknowledge the risks of over-reliance on simulated users and simulated judges, the possibility that future work may overfit to the benchmark’s specific feedback-generation assumptions, and the fact that benchmark conclusions could be taken as stronger evidence about “procedural memory” than the operational setup really justifies.

**Strengths And Weaknesses:**

**Strengths**
- The strongest contribution is the problem identification. The paper correctly observes that much of the recent LLM memory evaluation literature focuses on declarative retrieval, typically in long-context reading-comprehension-style settings, and does not directly test whether a system can improve from accumulated user feedback. This is an important gap, and the paper motivates it clearly.

- The benchmark itself is a substantial engineering contribution. The three-module design is clean and extensible, the off-policy and on-policy settings are sensible, and the overall dataset coverage is broad.

- The empirical negative findings are valuable. In particular, the results suggest that current memory-based systems do not clearly outperform simple retrieval baselines on this broader benchmark, and that some of them suffer from serious efficiency issues.

**Weaknesses**

- My main concern is that the conceptual framing is somewhat stronger than what the benchmark actually tests. The paper leans heavily on the distinction between declarative and procedural memory, drawing on neuroscience and IR-inspired framing. However, in the setup, both kinds of information are still mostly passed to current systems as textual memory entries or retrieval corpus. As a result, the benchmark demonstrates difficulty in feedback utilization, but it does not yet cleanly separate systems that truly process procedural memory differently from those that simply retrieve more text. I think the benchmark would be better positioned as an evaluation of feedback utilization and continual improvement, rather than as evidence for a deeper architectural memory taxonomy.

- The experimental conclusions are directionally convincing, but some of the quantitative evidence is not conclusive. Absolute differences between systems are often modest after aggregation, and the benchmark depends heavily on LLM-as-user simulation, LLM-as-judge metric aggregation for some datasets, and min-max or z-score normalization across heterogeneous tasks. These choices are understandable in a benchmark of this scope, but they introduce enough modeling assumptions that the strongest claims should be phrased carefully.

- Finally, the benchmark infrastructure appears rich enough to support more analysis than the paper currently provides. For example, I would have liked a more systematic study of feedback quantity, feedback type, and retrieval relevance. These would help explain the negative results.

Overall, I think this is a useful benchmark paper addressing an important evaluation gap. My reservations are mainly about the strength of the framing and the breadth of the conclusions, not about whether the benchmark itself is valuable.

---

> ### Author Rebuttal · Authors · 2026-03-27
>
> We thank the reviewer for the constructive feedback.
>
> ## Weakness 1: Conceptual Framing
>
> **Reviewer's Concern:** The paper leans on declarative/procedural memory distinction ...
>
> **Our Response:**
>
> To the best of our knowledge, there wasn't any methods proposed specifically for procedural memory. Most memory systems process both declarative and procedural information through similar textual interfaces.
>
> However, this doesn't affect the main contribution: MemoryBench is the first benchmark to provide and evaluate **procedural memory** (feedback logs) alongside declarative memory. As stated in Section 3.3: *"The feedback logs provided in MemoryBench are non-factual information describing how the system performed... Existing memory-based LLM systems simply treat all inputs as declarative memory... This limits their ability to utilize procedural memory."*
>
> The negative finding—that current systems fail to effectively utilize procedural memory—is itself an important contribution highlighting the need for new architectures in future.
>
> ---
>
> ## Weakness 2: Quantitative Evidence
>
> **Reviewer's Concern:** Absolute differences are modest; heavy reliance on LLM-as-user simulation...
>
> **Our Response:**
>
> We provide additional evidence in the Appendix:
>
> 1. **Statistical significance with variance**: We report standard deviations across test data in Table 13 & 14 (Appendix A.3.4), showing that differences are statistically meaningful.
>
> 2. **Per-dataset raw metrics:** Table 12 provides detailed results with original dataset-specific metrics before aggregation.
>
> 3. **Cross-normalization robustness:** Tables 10-11 show consistent rankings across min-max and Z-score normalization.
>
> 4. **Cross-backbone robustness:** Tables 15 & 22 show consistent rankings across Qwen3-32B and Mistral3.2-24B backbones.
>
>
> ---
>
> ## Weakness 3: Benchmark Could Support More Analysis
>
> **Reviewer's Concern:** More systematic study of feedback quantity, feedback type...
>
> **Our Response:**
>
> We conducted additional analyses in the Appendix:
>
> 1. **Feedback effect evaluation** (Tables 16-18): Feedback helps improve performance on most datasets.
>
> 2. **Stepwise off-policy experiments** (Figures 8-10): Performance evolution with accumulating feedback—each step feeds 100 training instances.
>
> 3. **On-policy experiments** (Section A.3.6, Figures 5-7): We conducted on-policy experiments showing real-time learning trajectories.
>
> 4. **Different feedback types** (Table 20): Action feedback (like/copy) shows similar patterns to verbal feedback.
>
> 5. **Case study** (Table 23): Detailed analysis of retrieved memories and responses, explaining why RAG outperforms memory systems.
>
>
> ---
>
> ## Key Question 1: Performance as Function of Feedback Volume
>
> We have provided the feedback volumne analysis in the stepwise results in Figures 8-10 for off-policy experiments (A.3.7). In each step, we feed 100 training instances to the system. Here are some key observations:
>
> - RAG systems (BM25-S, Embed-S) show relatively stable increasing performance with more feedback volumne.
> - Memory-based systems show more variance, indicating instability in utilizing feedback.
> - Performance patterns differ across domains (Open, Academic, Legal). Open is more stable, and Academic and Legal fluctuate more.
>
> The on-policy results (Figures 5-7) further show real-time learning trajectories, with systems evaluated after each batch of interactions.
>
> ---
>
> ## Key Question 2: Why Memory Systems Underperform RAG
>
> We provide several pieces of diagnostic evidence:
>
> 1. **Case study** (Table 23, Section A.3.11): We analyzed retrieved memories and responses. Memory-based systems retrieve entries with more noise (e.g., "Speaker", "User asked about") while RAG retrieves more relevant content.
>
> 2. **Time consumption analysis** (Section A.3.8, Table 19, Figure 4): Memory-based systems have significantly higher latency (Figure 11). Due to the efficiency issues, some memory baselines cannot provide responses within a reasonable time, so we have to ignore them in some parts of the experiments.
>
>
> As mentioned in Section 3.3, *"Existing memory-based LLM systems treat all inputs as declarative memory... This limits their ability to utilize procedural memory."* RAG's simpler retrieval produces less noisy context, highlighting the need for new architectures.
>
> ---
>
> ## Response to Limitations
>
> **Reviewer's Concern:** The paper should more directly acknowledge risks of over-reliance on simulated users and judges.
>
> **Our Response:**
>
> Thanks for the suggestion! We will add a limitations section.
>
> As an initial step to address this problem, we validated our simulator through human annotation (Table 7):
>
> - **Naturalness:** Kappa = 0.093 (indistinguishable)
> - **Relevance:** 59.5% win rate for simulated feedback
> - **Rejection p-value > 0.05**
>
> Cross-backbone validation (Table 22) shows consistent rankings across different simulators (Qwen3-32B vs. Mistral3.2-24B). Hope this could make the results more reliable.

---

> > ### Author Rebuttal · Reviewer_g9pS · 2026-04-05
> >
> > The rebuttal addressed my main concerns. I have updated my score.

---

> > > ### Author Response · Authors · 2026-04-07
> > >
> > > Thank you so much for you acknowledgment! Please do not hesitate to let us know if you have any further questions or need anything!

---

### Decision · Program_Chairs · 2026-04-30

**Decision:**

Accept (spotlight)

**Comment:**

The paper introduces MemoryBench, a benchmark for evaluating memory and continual learning in LLM systems. The central claim is that existing LLM memory benchmarks mostly test retrieval from long context or stored facts, but do not evaluate whether systems can improve over time by learning from user feedback. To address this, the paper proposes a benchmark framework with three components: a Task Provider that supplies queries, context, and evaluation metadata; a User Simulator that generates explicit and implicit feedback; and a Performance Monitor that evaluates downstream task performance. The benchmark covers 11 public datasets across three domains, four input/output format regimes, and two languages, and it distinguishes between declarative memory, episodic/semantic information, and procedural information derived from feedback logs.

The paper received high scores and all four reviewers agree that it should be accepted.